# Thermodynamic and dynamic ice thickness contributions in the Canadian Arctic Archipelago in NEMO-LIM2 numerical simulations

Xianmin Hu[1,+], Jingfan Sun[1,++], Ting On Chan[1,+++], and Paul G. Myers[1]

[1]Department of Earth and Atmospheric Sciences, University of Alberta, Edmonton, T6G 2E3, Canada
[+]now at Bedford Institute of Oceanography, Fisheries and Oceans Canada, Dartmouth, Nova Scotia, Canada
[++]summer intern from Zhejiang University, 38 Zheda Road, Hangzhou, China, 310027
[+++]now at Skytech Solutions Ltd., Canada

*Correspondence to:* Xianmin Hu(xianmin@ualberta.ca)

**Abstract.** Sea ice thickness evolution within the Canadian Arctic Archipelago (CAA) is of great interest to science, as well as local communities and their economy. In this study, based on the NEMO numerical frame work including the LIM2 sea ice module, simulations at both $1/4°$ and $1/12°$ horizontal resolution were conducted from 2002 to 2016. The model captures well the general spatial distribution of ice thickness in the CAA region, with very thick sea ice ($\sim 4\,m$ and thicker) in the northern CAA, thick sea ice ($2.5\,m$ to $3\,m$) in the west-central Parry Channel and M'Clintock Channel, and thin ($< 2\,m$) ice (in winter months) on the east side of CAA (e.g., eastern Parry Channel, Baffin Islands coast) and in the channels in southern areas. Even though the configurations still have resolution limitations in resolving the exact observation sites, simulated ice thickness compares reasonably (seasonal cycle and amplitudes) with weekly Environment and Climate Change Canada (ECCC) New Icethickness Program data at first-year landfast ice sites except at the northern sites with high-concentrations of old ice. At $1/4°$ to $1/12°$ scale, model resolution does not play a significant role in the sea ice simulation except to improve local dynamics because of better coastline representation. Sea ice growth is decomposed into thermodynamic and dynamic (including all non-thermodynamic processes in the model) contributions to study the ice thickness evolution. Relatively smaller thermodynamic contribution to ice growth between December and the following April is found in the thick and very thick ice regions, with larger contributions in the thin ice covered region. Wavelet analysis of the hourly simulated ice fields at two sites clearly shows the thermodynamic contribution has seasonal and diurnal cycles while only the seasonal cycle is significant for the total ice thickness. High frequency changes are found in both fields during the sea ice melting and formation process, particularly in the melting season. No significant variation in winter maximum ice volume is found in the northern CAA and Baffin Bay while a decline ($r^2 \approx 0.6, p < 0.01$) is simulated in Parry Channel region. The two main contributors (thermodynamic growth and lateral transport) balance each other with large inter-annual variability, but further quantitative evaluation is required.

## 1 Introduction

The Canadian Arctic Archipelago (CAA), the complex network of shallow-water channels adjacent to the Arctic ice pack, has been a scientific research hot spot for a long time. Scientifically, it is an important pathway delivering cold fresh Arctic

water downstream (e.g., Prinsenberg and Hamilton, 2005; Melling et al., 2008; Dickson et al., 2007; Peterson et al., 2012), that eventually feeds the North Atlantic Ocean, where the watermass formation and ocean dynamics play a key role in the large scale meridional overturning circulation (MOC) and global climate variability (e.g., Rhein et al., 2011; Hátún et al., 2005; Marshall et al., 2001; Vellinga and Wood, 2002). Economically, shipping through the CAA , via the Northwest Passage (NWP), is of particular interest to commercial transport between Europe and Asia because of the great distance savings compared to the current route through the Panama Canal (e.g., Howell et al., 2008; Pizzolato et al., 2016, 2014). This has been a hot topic under the context that Northern Hemisphere sea ice cover has been declining dramatically (e.g., Parkinson et al., 1999; Serreze et al., 2007; Parkinson and Cavalieri, 2008; Stroeve et al., 2008; Comiso et al., 2008; Parkinson and Comiso, 2013), especially after 2007. Besides the harsh weather and other safety issues (e.g., icebergs), the biggest concern for using the NWP is still the condition of sea ice, especially high concentrations of thick multiyear ice (MYI) (Melling, 2002; Howell et al., 2008; Haas and Howell, 2015).

Lietaer et al. (2008) estimated about 10% of the total Northern Hemisphere sea ice volume is stored within the CAA. Sea ice within the CAA region is a combination of both first-year ice (FYI) and MYI. MYI is both locally formed and imported from the Arctic Ocean, and normally located in the central-west Parry Channel and northern CAA (e.g., Melling, 2002; Howell et al., 2008, 2013). Since the late 1970s, the ice free season has extended by about one week per decade (Howell et al., 2009), with a statistically significant decrease of 8.7% per decade in the September FYI cover. Reduction in the September MYI cover is also found to be -6.4% per decade until 2008 (Howell et al., 2009). But this trend was not "yet statistically significant" due to the inflow of MYI from the Arctic Ocean mainly via the Queen Elizabeth Islands (QEI) gates in August to September (Howell et al., 2009). With extended data in recent years (until 2016), Mudryk et al. (2017) showed that the summer MYI decline rate has almost doubled. Even though the Arctic Ocean ice pack also extends to the CAA region through M'Clure Strait, the net sea ice flux is small and usually leads to an outflow from the CAA (Kwok, 2006; Agnew et al., 2008; Howell et al., 2013).

Although there is increasing demand for sea ice thickness information within this region, there are still very limited records available (Haas and Howell, 2015). Melling (2002) analyzed drill-hole data measured in winters during 1971–1980 within the Sverdrup Basin (the marine area between Parry Channel and QEI gates, see Fig. 1), and found sea ice in this region is landfast (100% concentration without motion) for more than half of the year (from October-November to late July) with a mean late winter thickness of 3.4 m. Sub-regional means of the ice thickness can reach 5.5 m, but very thick multiyear ice was found to be less common, which is likely due to the melting caused by tidally enhanced oceanic heat flux in this region (Melling, 2002). The seasonal transport of the old ice from the Sverdrup Basin down to the south was known to occur (Bailey, 1957). which helps to create another major region with severe MYI conditions in the CAA, the central Parry Channel and M'Clintock Channel (see Fig. 1 for the location). Based on two airborne electromagnetic (AEM) ice thickness surveys conducted in May 2011 and April 2015, Haas and Howell (2015) estimated the ice thickness to be 2 to 3 m in this region with MYI thicker than 3 m on average. This supports the general spatial distribution of ice thickness within the CAA, thicker in the north and relatively thinner in the south.

Observations were not only limited in time but also in spatial coverage, thus, numerical simulations are required to better understand the ice distribution and variability in the CAA (e.g., Dumas et al., 2006; Sou and Flato, 2009; Hu and Myers, 2014). Dumas et al. (2006) evaluated the simulated ice thickness at CAA meteorological stations but with a uncoupled one dimensional (1D) sea ice model (Flato and Brown, 1996). The variability and trends of landfast ice thickness within the CAA were systematically studied by a recent paper from Howell et al. (2016) based on historical records at observed sites (Cambridge Bay, Resolute, Eureka and Alert) and numerical model simulations over the 1957–2014 period. They found statistically significant thinning at the sites except at Resolute, and the detrended inter-annual variability is high (negative) correlated with snow depth due to the insulating effect of the snow (Brown and Cote, 1992). Although some of the numerical simulations used in Howell et al. (2016) produced a reasonable seasonal cycle, generally, these simulations overestimated ice thickness and did not do a good job in capturing the trend. In addition, the lack of horizontal resolution in these models were also pointed out in Howell et al. (2016).

In this paper, we focus on the simulated CAA sea ice thickness over recent years (2002–2016). First, to evaluate the skill of a numerical model in simulating sea ice thickness, comparisons are done between the simulations and landfast ice thickness at several sites in the CAA. Two features of the simulations will then be discussed: 1) the relative importance of thermodynamic and dynamic processes in sea ice growth/melting in simulating the possible variability of sea ice and ocean surface fields in the CAA; 2) high frequency changes in ice growth/melting processes. This paper starts with a brief description of the numerical simulations and observational data used in this study. Then the evaluation of simulated ice thickness in the CAA region is presented in section 3.1. The spatial distribution, temporal evolution (at selected sites) of thermodynamic and dynamic ice thickness contributions, as well as the high frequency (up to diurnal cycle) variability in the two components, are studies in section 3.2. Ice volume budgets in the northern CAA, Parry Channel and Baffin Bay are discussed in section 3.3. Concluding remarks and discussions are given in section 4.

## 2  Method and Data

### 2.1  Numerical model setup

In this study, the coupled ocean sea ice model, the Nucleus for European Modeling of the Ocean (NEMO, available at https://www.nemo-ocean.eu) version 3.4 (Madec and the NEMO team, 2008), is utilized to conduct the numerical simulations. The model domain covers the Arctic and the Northern Hemisphere Atlantic (ANHA) with two open boundaries, one close to Bering Strait in the Pacific Ocean and the other one at $20°$S across the Atlantic Ocean (Fig. 1, inset). The model mesh is extracted from the the global tripolar grid, ORCA (Drakkar Group, 2007) with two different horizontal resolutions, $1/4°$ (hereafter ANHA4) and $1/12°$ (hereafter ANHA12). The highest horizontal resolution is $\sim 2\,km$ for ANHA12 and $\sim 6\,km$ for ANHA4 in Coronation Gulf–Dease Strait region, which is near the artificial pole over northern Canada (Fig. 1), and the lowest resolution is $\sim 9\,km$ for ANHA12 and $\sim 28\,km$ for ANHA4 at the equator. In the vertical, there are 50 geopotential levels with high resolution focused in the upper ocean. Layer thickness smoothly transitions from $\sim 1m$ at surface (22 levels for the top $100\,m$) to $458\,m$ at the last level.

The sea ice module used here is the Louvain la-neauve Ice Model Version 2 (LIM2) with an elastic-viscous-plastic (EVP) rheology (Hunke and Dukowicz, 1997), including both thermodynamic and dynamic components (Fichefet and Maqueda, 1997). It is based on a three-layer ( one snow layer and two ice layers of equal thickness) model proposed by Semtner Jr (1976) with two ice thickness categories (mean thickness and open water). The sea ice module is coupled to the ocean module every model step. The elastic time scale is tuned small enough to damp the elastic wave in the EVP approach (see Table 1), based on the discussions in Hunke and Dukowicz (1997). Note that recent studies (e.g., Lemieux et al., 2012; Bouillon et al., 2013; Williams et al., 2017) showed that more iterations are needed to reach a viscous-plastic (VP) solution. Without doing that, the divergence field will be affected, i.e., being noisy (Dupont, 2017). Thus, to what degree it will impact the final averaged ice thickness will vary in space. Such an investigation in the CAA is beyond the scope of this study. A no-slip boundary condition is applied for sea ice in the simulations, which means the ice can have zero velocity along the coast. However, it should be noted that the sea ice module used in this study does not include a representation of landfast ice (e.g., Lemieux et al., 2016), which may negatively impact the sea ice simulation where landfast ice exists.

**Table 1.** Sea ice module parameters used in our simulations

| parameter | ANHA4 | ANHA12 |
|---|---|---|
| time step (seconds) | 1080 | 180 |
| subcycling iterations | 150 | 120 |
| timescale of elastic wave (seconds) | 320 | 60 |

Two simulations, ANHA4-CGRF and ANHA12-CGRF, are integrated from January 1st 2002 to December 31 2016. The initial conditions, including three dimensional (3D) ocean fields (temperature, salinity, zonal velocity and meridional velocity) as well as two dimensional (2D) sea surface height (SSH) and sea ice fields (concentration and thickness) are taken from

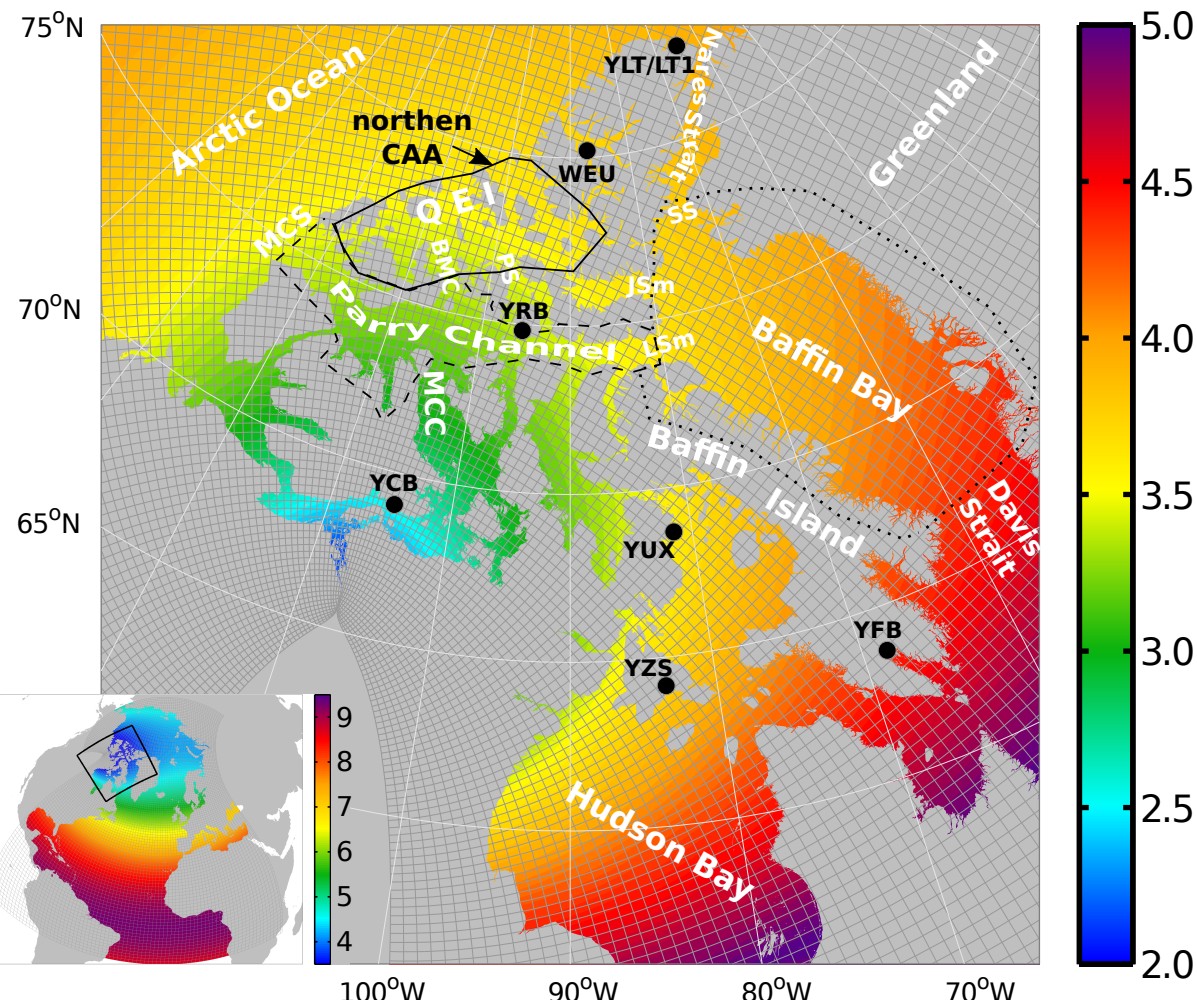

**Figure 1.** ANHA12 (inset) model mesh (every 10th grid point) and horizontal resolution (colours, unit: kilometers) in the Canadian Arctic Archipelago (QEI: Queen Elizabeth Islands; MCS: M'Clure Strait; MCC: M'Clintock Channel; BMC: Byam Martin Channel; PS: Penny Strait; JSm: Jones Sound Mouth; LSm: Lancaster Sound Mouth; SS: Smith Sound) and Hudson Bay region (thick black box highlighted in the inset). Note the colour scale is different from that used in the inset). Ice thickness observation sites (YZS: Coral Harbour, YUX: Hall Beach; YFB: Iqaluit; YCB: Cambridge Bay; YRB: Resolute; WEU: Eureka; YLT: Alert; LT1: Alert LT1) are shown with black circles on the map. Detailed location information of observation sites is available in table 2.

from the Global Ocean Reanalysis and Simulations (GLORYS2v3) produced by Mercator Ocean (Masina et al., 2015). Open boundary conditions (temperature, salinity and horizontal ocean velocities) are derived from the monthly GLORYS2v3 product as well. At the surface, the model is driven with high temporal (hourly) and spatial resolution ($33\,km$) atmospheric forcing data provided by Canadian Meteorological Centre (CMC) Global Deterministic Prediction System (GDPS) ReForecasts (CGRF)
5 dataset (Smith et al., 2014), including 10 m wind, 2 m air temperature and humidity, downwelling and longwave radiation

flux, and total precipitation. These forcing fields are linearly interpolated onto the model grid. Inter-annual monthly $1° \times 1°$ river discharge data from Dai et al. (2009) as well as Greenland meltwater ($5\,km \times 5\,km$) provided by Bamber et al. (2012) is carefully (volume conserved) remapped onto the model grid.

With the same setting as ANHA4-CGRF but driven with the inter-annual atmospheric forcings from the Coordinated Ocean-ice Reference Experiments version 2 (CORE-II) (Large and Yeager, 2009), another ANHA4 simulation, ANHA4-CORE, integrated from January 1st 2002 to December 31 2009, is also conducted to study the sensitivity of the sea ice simulation to the atmospheric forcings. The CORE-II provides fields at various temporal resolutions, a) 6-hourly 10-m surface wind, 10-m air temperature and specific humidity; b) daily downward longwave and shortwave radiation; c) monthly total precipitation and snowfall.

No temperature or salinity restoring is applied in any of the simulations used in this study. Without such constraints the model evolves freely in time to help understand better the limitations of the physical processes represented by the model.

## 2.2 Environment and Climate Change Canada New Arctic Ice Thickness Program

To evaluate the performance of the model in terms of ice thickness, simulated ice thickness is compared to the observed landfast ice data from Environment and Climate Change Canada (ECCC) New Icethickness Program (hereafter ECCC thickness). The new ECCC thickness program, the second stage of the Original Ice Thickness Program Collection used in Dumas et al. (2006), started in the fall of 2002 , and continued to the present at only 11 stations (including sites on lakes). Measurements were conducted weekly at approximately the same location close to shore between freeze-up and break-up (when the ice was safe to walk on) with a special auger kit or a hot wire ice thickness gauge. Note the measurement represents the immobile level first-year (seasonal) ice of uniform thickness that forms close to shore, however, simulated ice thickness, e.g., due to resolution, generally, is an estimation of the mean state of different types of ice (e.g., first-year level ice, young ice and old ice).

Data is made available to the public by Environment and Climate Change Canada under the Open Government License (Canada) on http://open.canada.ca/data/en/dataset. Eight coastal sites, Coral Harbour, Hall Beach, Iqaluit, Cambridge Bay, Resolute, Eureka, Alert and Alert LT1, were used in this study. The remaining three sites are on lakes (not included in our simulations). The detailed location information of each site can be found in Fig. 1 and Table2.

Unlike the 1D model used in Dumas et al. (2006), which can be applied at the exact location where the measurements were carried out, three dimensional (3D) models usually have horizontal resolution issues in resolving the observation sites, even with the high resolution ANHA12 configuration used in this study. Interpolation is needed to do the comparison between simulated fields and observations. This is also mentioned in Howell et al. (2016). To interpolate simulated fields onto the nearest water point $(x_k, y_k)$ of each observation site $(x_{obs}, y_{obs})$, we utilized a modified inverse distance weighting (IDW) method (eq. (1)) proposed by Renka (1988):

**Table 2.** ECCC ice thickness station locations (only sites used in this study)

| site | longitude | latitude |
|---|---|---|
| Coral Harbour | $83.153°W$ | $64.130°N$ |
| Hall Beach | $81.230°W$ | $68.780°N$ |
| Iqaluit | $68.517°W$ | $63.726°N$ |
| Cambridge Bay | $105.06°W$ | $69.113°N$ |
| Resolute | $94.884°W$ | $74.684°N$ |
| Eureka | $85.942°W$ | $79.986°N$ |
| Alert LT1 | $62.593°W$ | $82.602°N$ |
| Alert | $62.420°W$ | $82.753°N$ |

$$f_i = [\frac{R_w - d_k}{R_w \, d_k}]^2 \tag{1a}$$

$$w_i = \frac{f_i}{\sum_{i=1}^{N} f_i} \tag{1b}$$

$$Q_{target} = \sum_{i=1}^{N} Q_i \, w_i \tag{1c}$$

where $R_w$ is the influence radius about point $(x_k, y_k)$, $d_k$ is the distance from point $(x_k, y_k)$ to each neighboring point $(x_i, y_i)$,
$f_i$ is the inverse distance function, $w_i$ is the weight function on each neighboring point $(x_i, y_i)$, $N$ is the number of neighboring points within $R_w$, $Q_i$ is variable value on each neighboring point and $Q_{target}$ is the final result. In practice, nine neighboring points, including point $(x_k, y_k)$, were considered in the calculation. As $R_w$ is set to the maximum value of $d_k$ and land points should be excluded, eventually up to eight effective points are used in our interpolation process.

## 2.3 Wavelet analysis

Constrained by sampling frequency in observations and atmospheric forcing data, very few modelling studies have ever studied ice thickness variations on a time scale shorter than monthly. Using a 1D thermodynamic sea ice model, Hanesiak et al. (1999) found the hourly atmospheric forcing, which resolves the diurnal cycle, can produce more realistic results in terms of sea ice melting processes (i.e., simulated breakup dates, open water duration and snow ablation) compared to the non-diurnal-resolved forcing (e.g., daily). The significant differences is caused by the nonlinearities in surface energy balance that affects the snow
depth, albedo and surface temperature (Hanesiak et al., 1999). Thus, in this study, more realistic sea ice melting and freezing processes are expected, given that the CGRF dataset provides hourly surface atmospheric forcing fields. Hourly simulated sea ice fields (ice thickness, thermodynamic ice production) are saved from the ANHA12-CGRF simulation between 2008 and 2016 to further study the high frequency processes in ice thickness evolution. Wavelet analysis (Torrence and Compo, 1998) is utilized to show both the periods of the ice thickness variation and their temporal evolution, The wavelet toolbox is accessible

from http://paos.colorado.edu/research/wavelets, and a bias correction discussed in Veleda et al. (2012) is also included in our analysis.

## 3 Results

In this section, first, the ice thickness reproduction ability within the CAA of the NEMO LIM2 configurations used in this study is examined via comparisons with the ECCC thickness. After that, the detailed thermodynamic and dynamic ice thickness changes, both the spatial distribution and temporal evolution at selected sites (Cambridge Bay and Resolute), based on the simulation outputs will be presented. Then follows the high frequency ice growth/melting processes at Cambridge Bay and Resolute. Ice volume balance, focusing on the thermodynamics contribution and lateral transport, in the northern CAA, Parry Channel and Baffin Bay will also be included at the end.

### 3.1 Ice thickness comparison

Figure 2 shows the ice thickness comparison with observations. In general, both ANHA4-CGRF (blue lines) and ANHA12-CGRF (red lines) simulations produce similar seasonal and inter-annual variations in ice thickness, which compare reasonably well at some sites (i.e., Cambridge Bay, Coral Harbour, Hall Beach, Resolute and Iqaluit) but not at the rest (Eureka, Alert and Alert LT1). The sites where the model produced much thicker ice are likely where significant concentration of old ice exists (CIS, 2011). Although the observations are missing in the sea ice melting season, an asymmetric seasonal cycle (a shorter faster melting period follows a relatively longer slow growth period), is evidenced by the available data, and reproduced by the simulations. This is clearly shown in the ice thickness seasonal cycle plot (Fig. 3). Taking account of the model resolution, the interpolated simulated ice thickness reflects actually the variability some distance off the coast rather than the exact observation locations. The geographic location differences, which is also related to model resolution, could also lead to discrepancies in the comparisons here. Thus, if the model can capture the seasonal cycle (e.g., multiple data points in both ice growth and melting seasons), the model is likely capable of simulating the process.

At Iqaluit, the model does a good job in most years during the initial ice growth period but failed to catch the thick sea ice in the next April and May (Fig. 3), particularly in 2014, 2015 and 2016 (Fig. 2h). This could be a local atmospheric forcing field bias, or a model resolution issue, i.e., the measurements captured very localized extremes beyond the ability of model to resolve. Similar behavior happens at Coral Harbour and Hall Beach (Fig. 2c and d). Further investigation is needed.

At Eureka, Alert and Alert LT1 sites (Fig. 2 and 3, e, f, and g), there are clear differences between the simulated ice thickness and the observations ($\sim 2\,m$ at Alert/Alert LT1 and $\sim 1\,m$ at Eureka). Note neither ANHA4 or ANHA12 has the capability to resolve the difference between Alert and Alert LT1, thus, the same simulated values are shown on the figure for both sites. The differences between simulations and observations could be an initial value problem, particularly at Eureka (Fig. 2g). However, given high concentrations of old ice are at these sites, observations represent the immobile level first-year ice only. Thus, the model and the observations may not be representing the some type of ice. At Alert/Alert LT1, both ANHA4-CGRF (blue line) and ANHA12-CGRF (red line) show similar inter-annual trends to that in GLORYS2v3 (which extends back to 1993, green line), meaning it is likely a pure initial value problem rather than the model equilibrium issue mentioned in Howell et al. (2016). In addition, the seasonal cycle is not clear in the GLORYS2v3 product. The issue is also present in some years, i.e., 2005–2007, in the ANHA4-CGRF and ANHA12-CGRF simulations. ANHA4-CORE (orange line) is generally improved compared with

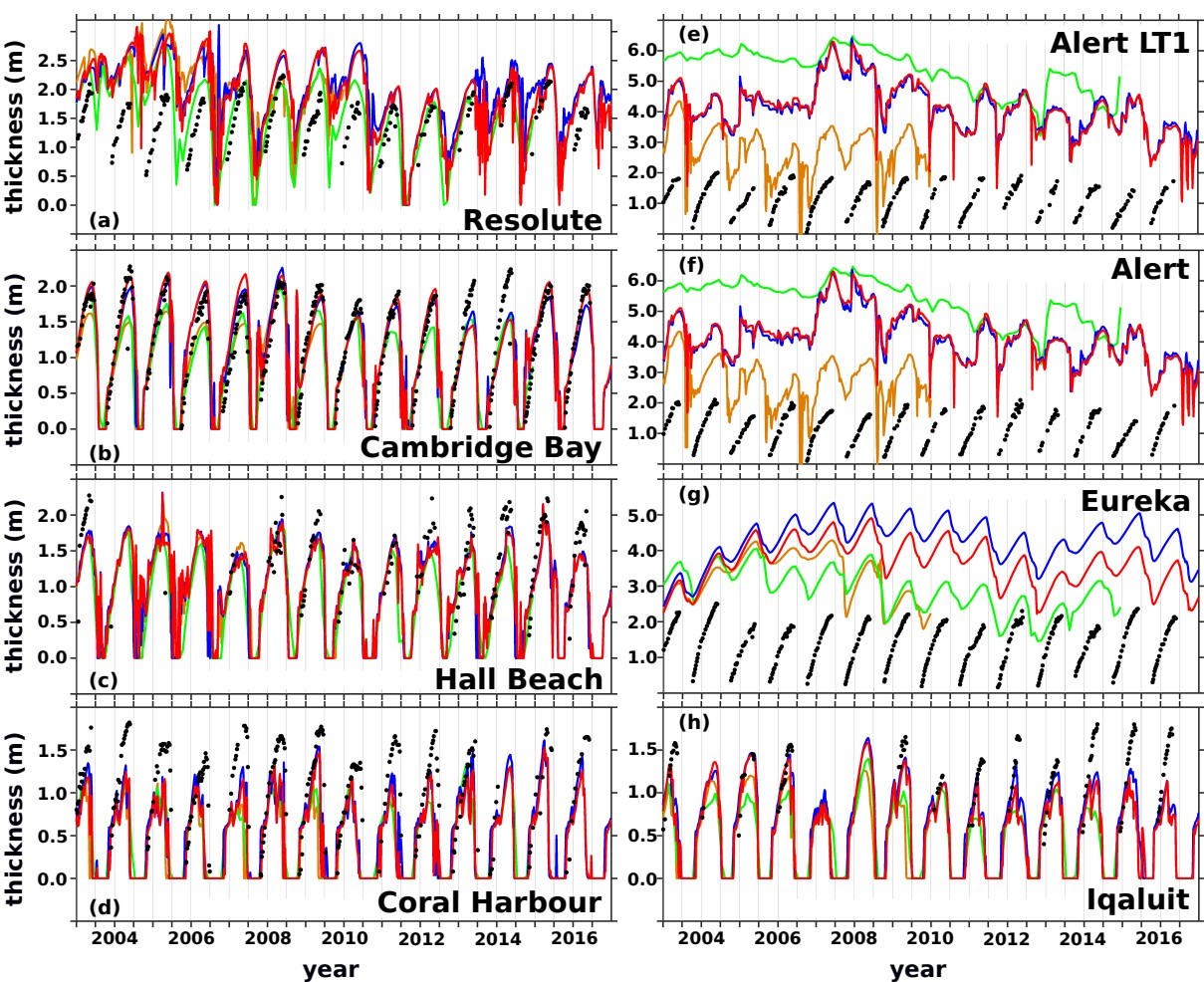

**Figure 2.** Simulated ice thickness at each selected ECCC thickness site (figure 1 and table 2, unit: meters) from January 2003 to December 2016 (orange: ANHA4-CORE simulation; green: GLORYS2v3 product; blue: ANHA4-CGRF simulation; red: ANHA12-CGRF simulation) against weekly ECCC observations (black dots). Note the GLORYS2v3 product is a monthly mean field while the rest of the simulations use 5-day averages. Different y-axis scales are used.

the observations in both the amplitude and seasonal cycle. However, this improvement was achieved by accident, and is related to a snow depth issue in this simulation (more details will be in the discussion session). Thus, it does not indicate that CORE-II forcing is performing better than other atmospheric forcing datasets in this region. The equilibrium issue, i.e., ice thickness keeping increasing, might happen at Eureka in our simulations with either CGRF or CORE-II forcing. The upward trend over 2005 to 2007 is also present in the observations but is missing in the GLORYS2v3 although GLORYS2v3 has a small thickness (which is likely due to data assimilation in GLORYS2v3 or an atmospheric forcing issue in 2005). Its trend does not reflect the real change/variability.

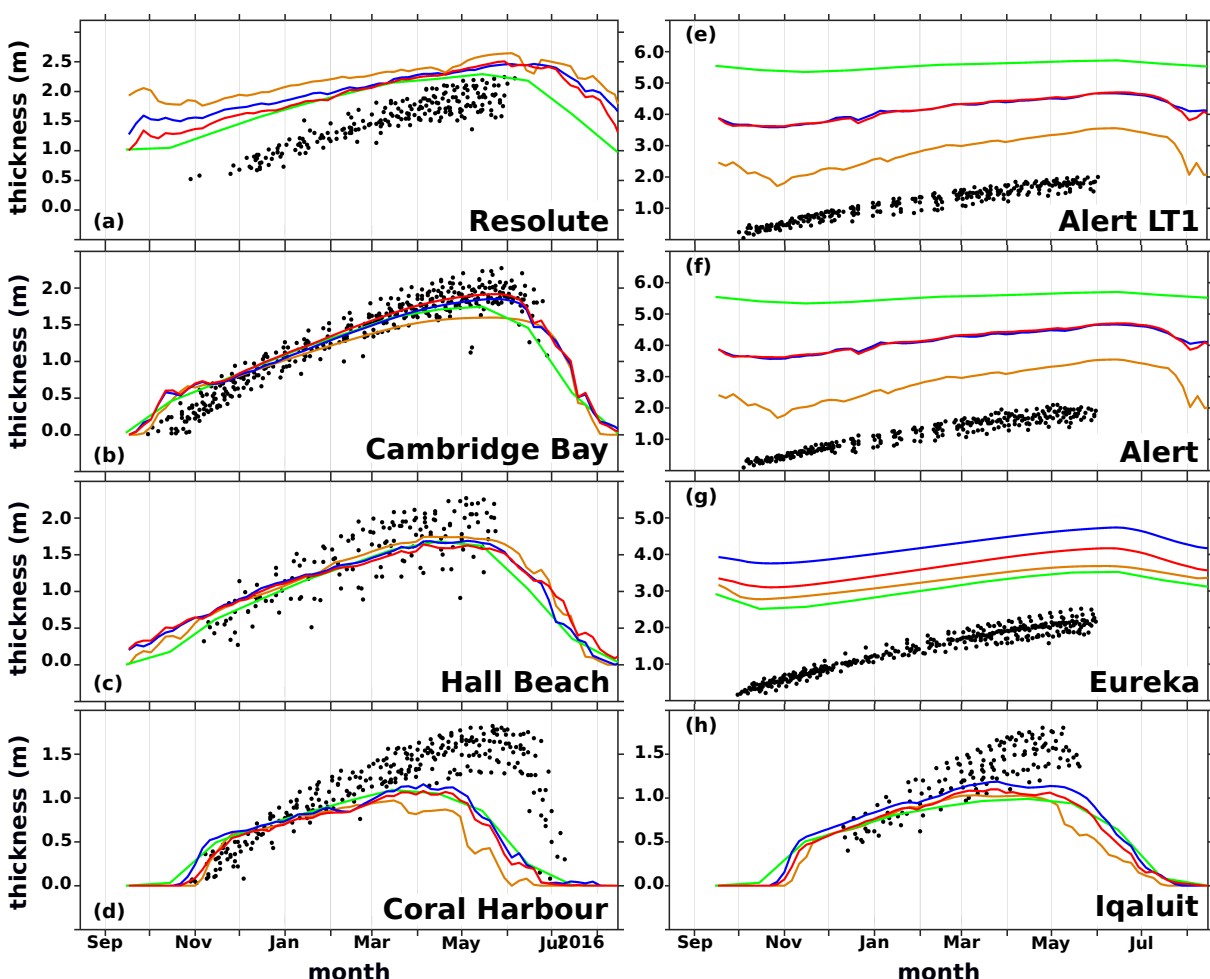

**Figure 3.** Similar to Fig. 2 but for ice thickness seasonal cycle (starting from September 17 to the next September 12, averaged over 2003 to 2016; ANHA4-CORE ends by 2009). Note observations are not averaged over time because the sampling time is different from year to year).

At Cambridge Bay, simulations (red and blue lines in Fig. 2b) with CGRF forcing show very good agreement with the observations except during the winters of 2013 and 2014. Considering the horizontal resolution of our simulations is not capable of resolving the inner bay at Cambridge Bay, the match in ice thickness between the simulation and observations indicates the variation of ice thickness within the inner Bay might be small. Both ANHA4-CORE and GLORYS2v3 simulations

5 underestimated the maximum values in winters by $\sim 0.5\,m$. This indicates CGRF forcing might provide more realistic surface inputs in this region.

At Resolute, it is more complicated (Fig. 2a). Prior to the significant sea ice melting in 2007, none of the simulations show ice free conditions in this region in summer. GLORYS2v3 shows relatively thinner ice in summer months but it is still $0.5\,m$ to

$1.5\,m$ thick. It could be the initial value problem. However, high frequently variations even in winter in the ANHA simulations suggest that the ice growth process is not dominated by a smoothly changing physical process (e.g., air temperature). Thus, it is likely due to another physical process such as advection from surrounding areas. This will be discussed more in the following section. Post 2007, the seasonal cycle in the sea ice field is more distinct, although ice free summer conditions do not happen every year. After 2010, simulations produce winter sea ice thickness much close to the observations.

## 3.2 Thermodynamic and dynamic ice thickness change

In the real world, both the thermodynamic and dynamic ice thickness processes are coupled together (occurring at the same time). However, with the assistance of numerical model, the two processes can be decoupled (shown in equation (2)) to better understand the relative importance of each process.

$$\Delta H_{total} = \Delta H_{thermal} + \Delta H_{dynamic} \tag{2}$$

where $\Delta H_{total}$ is the total ice thickness change over a specific time interval; $\Delta H_{thermal}$ is the ice thickness change due to vertical heat fluxes (through the atmosphere-ice-ocean interfaces); $\Delta H_{dynamic}$ is the ice thickness change due to dynamic processes. In practice, a simple approach is utilized to compute the two terms on the right side. $\Delta H_{thermal}$ is calculated based on the model thermal ice production. $\Delta H_{dynamic}$ is taken as the residual from the $\Delta H_{total}$.

### 3.2.1 Spatial distribution

Here we focus on ice growth process between December and April of the following year. Figure 4a and 4b show the simulated ice thickness in ANHA12 at the beginning of December and at the end of April, respectively. Geographically, at the end of April, a) very thick sea ice is located in the northern CAA ($\sim 4\,m$ by the end of April) with regional maximum ($> 4.5\,m$) at the openings to the Arctic Ocean. This is consistent with the ICESat and Cryosat-2 estimations (e.g., Laxon et al., 2013; Tilling et al., 2015; Kwok and Cunningham, 2015). b) less thick sea ice covers western, and central Parry Channel (just to the west of the site Resolute) and M'Clintock Channel with a thickness of $2.5\,m$ to $3\,m$. These values are similar to previous observations from airborne electromagnetic surveys (Haas and Howell, 2015) and satellite (Tilling et al., 2017). c) relatively thin ice ($< 2\,m$) is mainly in the southern CAA, eastern Parry Channel, coasts of Baffin Islands and within Hudson Bay. Invasion of the Arctic Ocean ice pack through the northern CAA openings and the advection from there into central Parry Channel are clearly shown in the figures, consistent with previous studies (e.g., Melling, 2002; Howell et al., 2008; Haas and Howell, 2015).

During the winter, sea ice grows everywhere in the CAA regions due to the thermodynamic cooling (Fig. 4c). But the total increase over the winter is not evenly distributed in space. Nor is ice growth largest in the north. Large thermodynamic ice growth is seen in the eastern CAA (eastern Parry Channel, Nares Strait, Baffin Island coast and western Hudson Bay), Amundsen Bay and many coastal regions (e.g., western coast of Banks Island, northern coast of western Parry Channel). Regions covered by thick sea ice (i.e., northern CAA, west-central Parry Channel and M'Clintock Channel) show less thermodynamic ice growth over the winter. This is particularly true in the northern CAA, likely due to the existence of already thick ice reducing the heat exchange between the ocean and atmosphere.

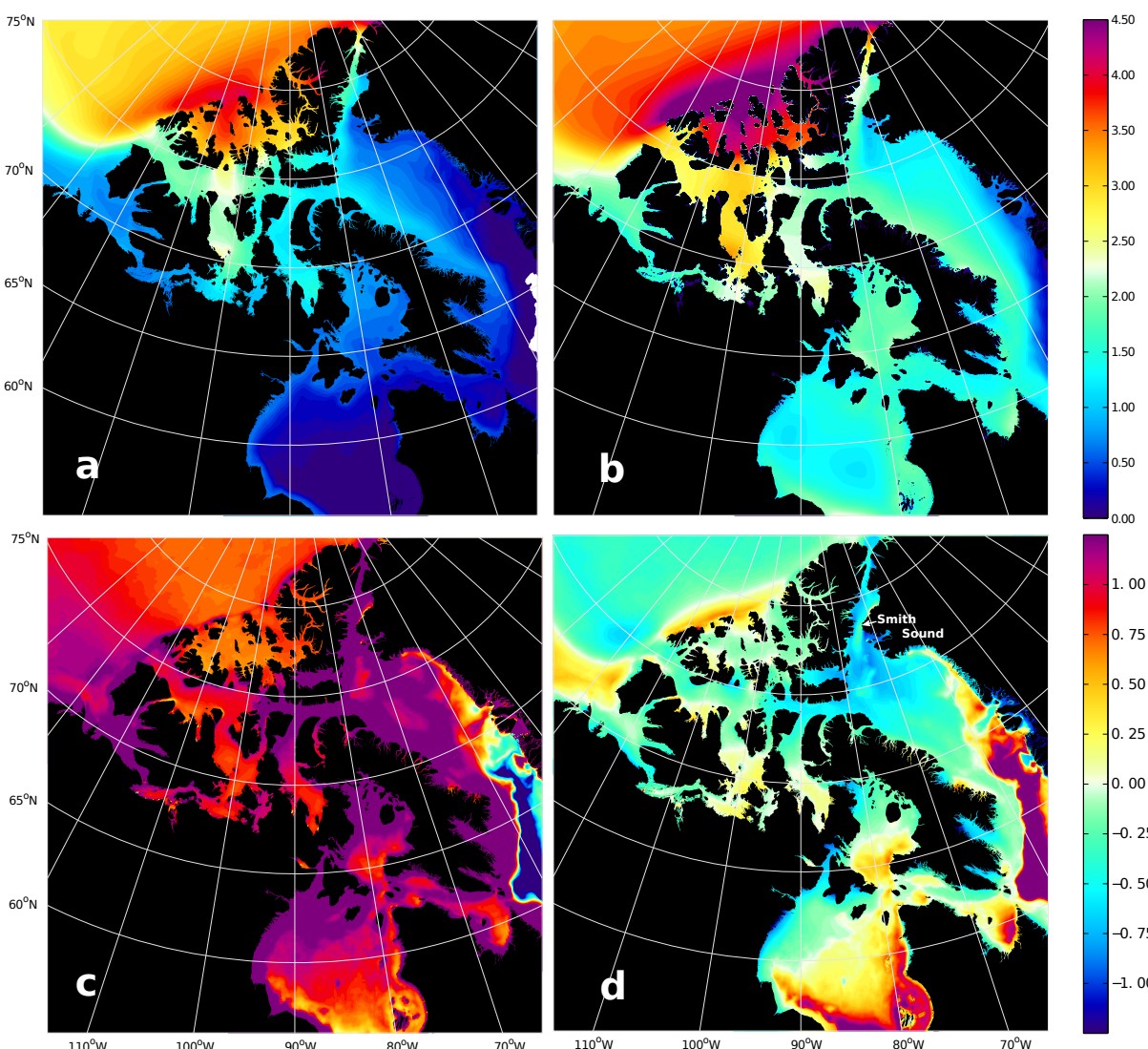

**Figure 4.** Upper panel shows the thickness (unit: meters) averaged over 2003–2016 at the beginning of December (a) and at end of April (b). Lower panel shows the thermodynamics component (c) and dynamic component ice thickness contribution (unit: meters) between December (a) and the following April (b) averaged over 2003-2015. ANHA12-CGRF simulation is used here.

The dynamic contribution to sea ice thickness is mainly negative (reduces local ice thickness) within the CAA (Fig. 4d). Large positive values ($0.4$ to $0.7\,m$) are shown along the Arctic Ocean coast off the CAA and within the Beaufort Sea. This is consistent with known sea ice convergence or strong advection of thick ice from upstream regions (Kwok, 2015; Maslanik et al., 2011). Within the northern CAA, west-central Parry Channel and M'Clintock Channel, there is $\sim 0.25\,m$ thick ice loss locally due to the dynamics. Note the positive values occurring in the south of M'Clintock Channel, suggesting a net convergence

there which contributes to the local ice thickening in winter. In the eastern CAA (e.g., eastern Parry Channel, Nares Strait and northwest corner of Baffin Bay), there are large negative dynamic thickness contributions, implying strong ice advection.

Although the North Water (NOW) Polynya (e.g., Dunbar, 1969; Melling et al., 2001) region is still ice covered by the end of April (Fig. 4b), the spatial distribution of negative dynamic ice thickness (which helps to remove local ice) captures the shape of NOW Polynya well. Weaker advection of sea ice at Smith Sound and to its south, which is likely to be caused by ice jamming, is also simulated by the model.

### 3.2.2 Seasonal cycle at Cambridge Bay and Resolute

Two sites, Cambridge Bay and Resolute, were selected to further study the seasonal cycle of the thermodynamic and dynamic ice thickness changes.

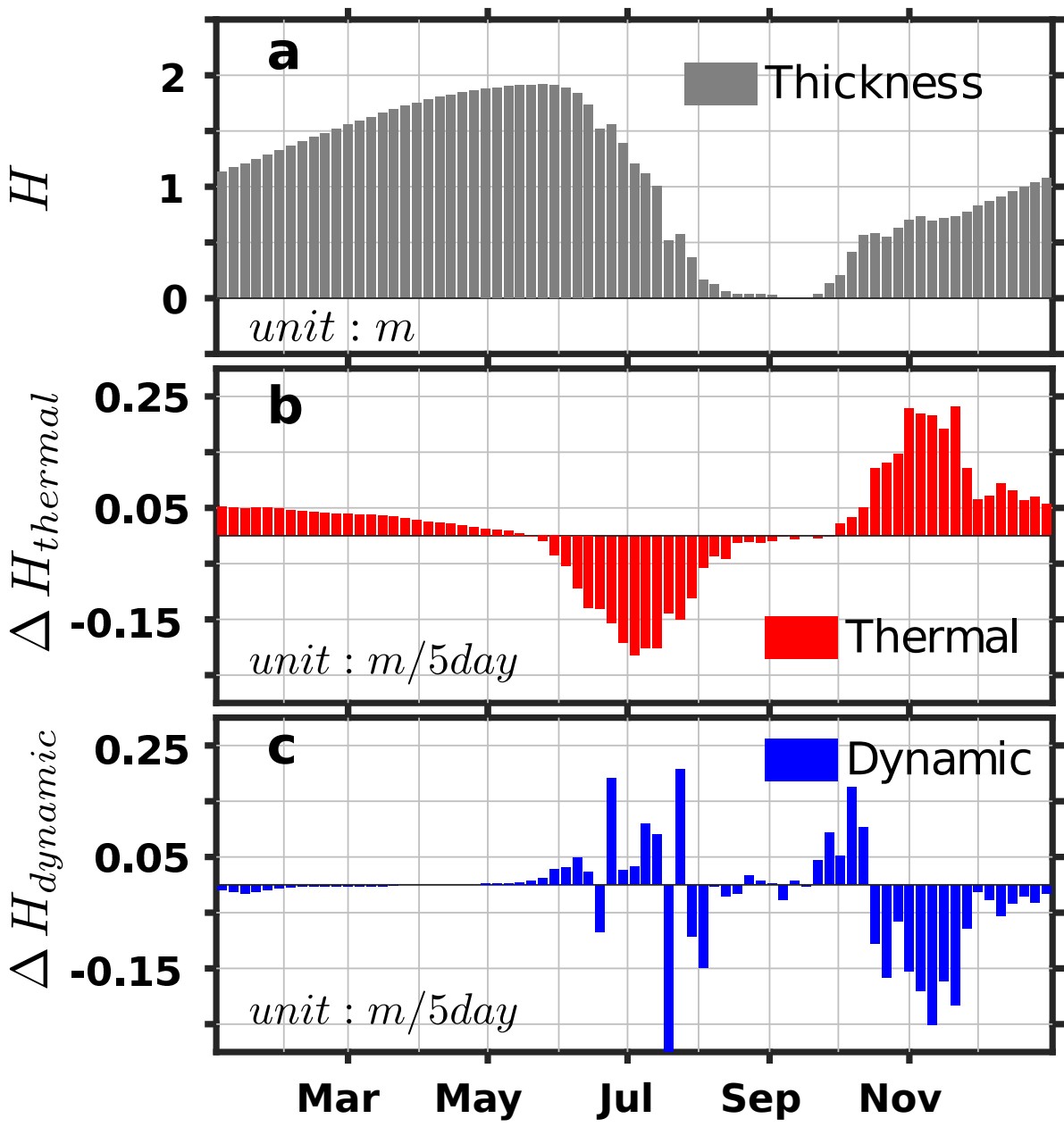

**Figure 5.** Seasonal cycle (averaged over 2003 to 2016) of ice thickness (a, unit:meters), dynamic (b) and thermodynamic (c) ice thickness changes (unit: meters per 5-day) at Cambridge Bay from the ANHA12-CGRF simulation. Note each x-grid line indicates the beginning of each month.

Figure 5 shows the seasonal cycle of ice thickness, 5-day $\Delta H_{dynamic}$ and $\Delta H_{thermal}$ averaged between 2003 and 2016 at Cambridge Bay. Sea ice reaches its maximal thickness ($\sim 2\,m$) in late May with ice free conditions for about two months (August and September). As the sea ice starts to form (October and November), both thermodynamics (e.g., due to cold temperature) and dynamics (e.g., local advection) play a role in the production of the ice thickness although with opposite contributions (Fig. 5b and c). Starting from December through to the end of the next May, it is almost a pure thermodynamic process that controls the ice thickness change. Note the thermodynamic ice production is not constant in time, it is about three times larger in the first period ($\sim 0.03\,m\,per\,day$) than in the later period ($\sim 0.01\,m\,per\,day$). The steady thermodynamic growth in the second period contributes to about half of the total ice thickness. During the ice melting period (June and July), the thermodynamics is the major player as well (Fig. 5c).

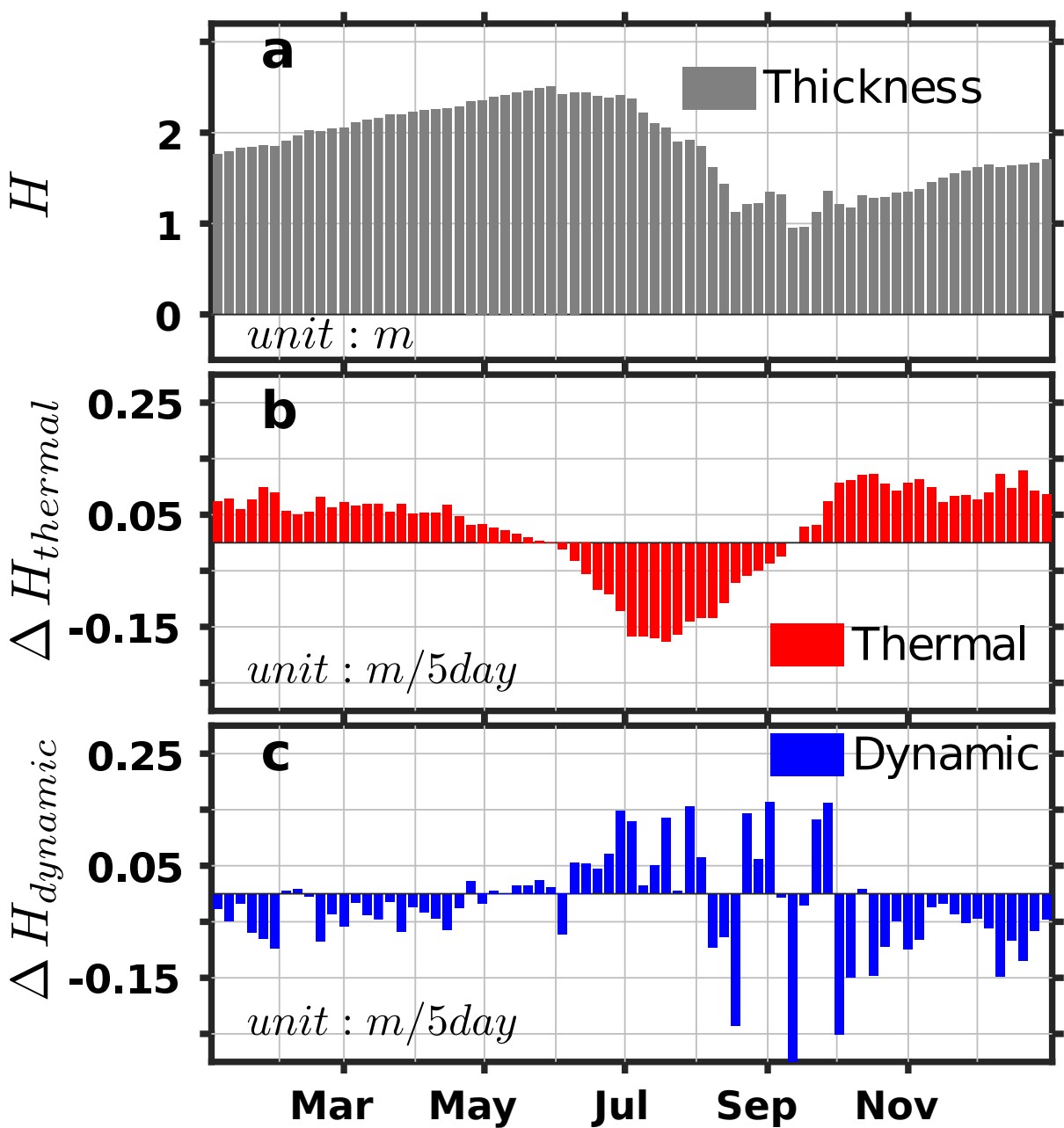

**Figure 6.** Same as Fig. 5 but at Resolute.

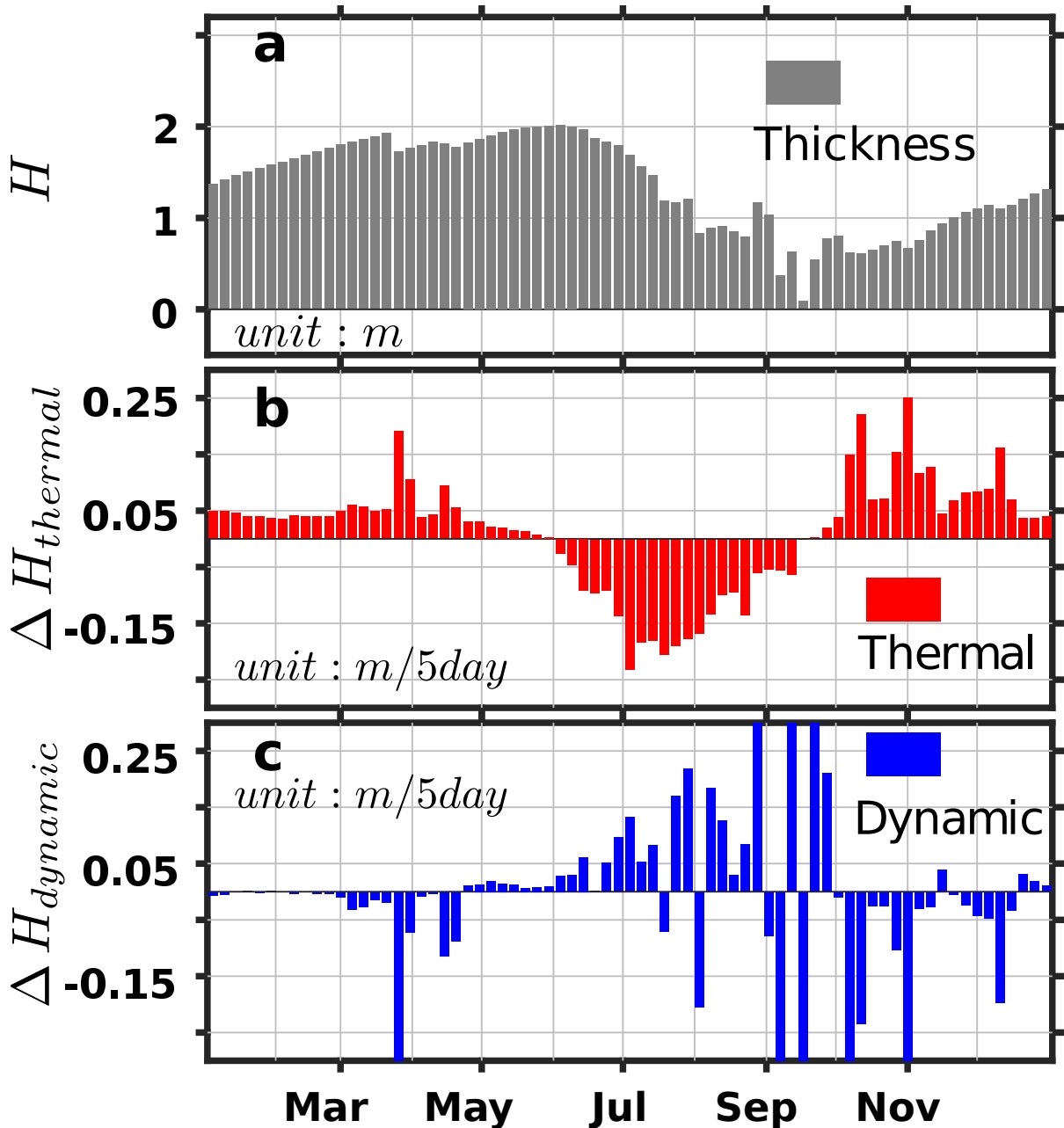

**Figure 7.** Same as Fig. 6 but only considering 2012 for ANHA12-CGRF.

At Resolute, on average, there is no ice free period ($\sim 2.5\,m$ at the end of May and $\sim 1\,m$ in August and September) (Fig. 6a), albeit with large inter-annual variability (Fig. 2a). For example, in 2012, there is an ice free period in the mid of September (Fig. 7a). The freeze-up date is about half a month earlier at Resolute than that at Cambridge Bay. The ice production is a little larger at the beginning (October to December), $\sim 0.02\,m\,per\,day$, than later (6b), $\sim 0.01\,m\,per\,day$, but the difference is not as noticeable as at Cambridge Bay (Fig. 5b). The relatively faster thermal growth lasts longer at Resolute than that at Cambridge Bay, likely due to local advection. These features are also applicable to a specific year, e.g., 2012 (Fig. 7). The non-thermodynamic contribution is more significant than at Resolute (Fig. 5c) but basically plays a negative role, i.e., slowing ice thickness increase during the winter season. Similarly to Cambridge Bay, the thermodynamics is the dominant factor melting the sea ice, with a melting peak in July. During the melting season, more ice can be advected to Resolute and melts later locally (Fig. 6c) than that at Cambridge (Fig. 5c).

### 3.2.3  High frequency thermodynamic processes during ice formation and melting at Cambridge Bay and Resolute

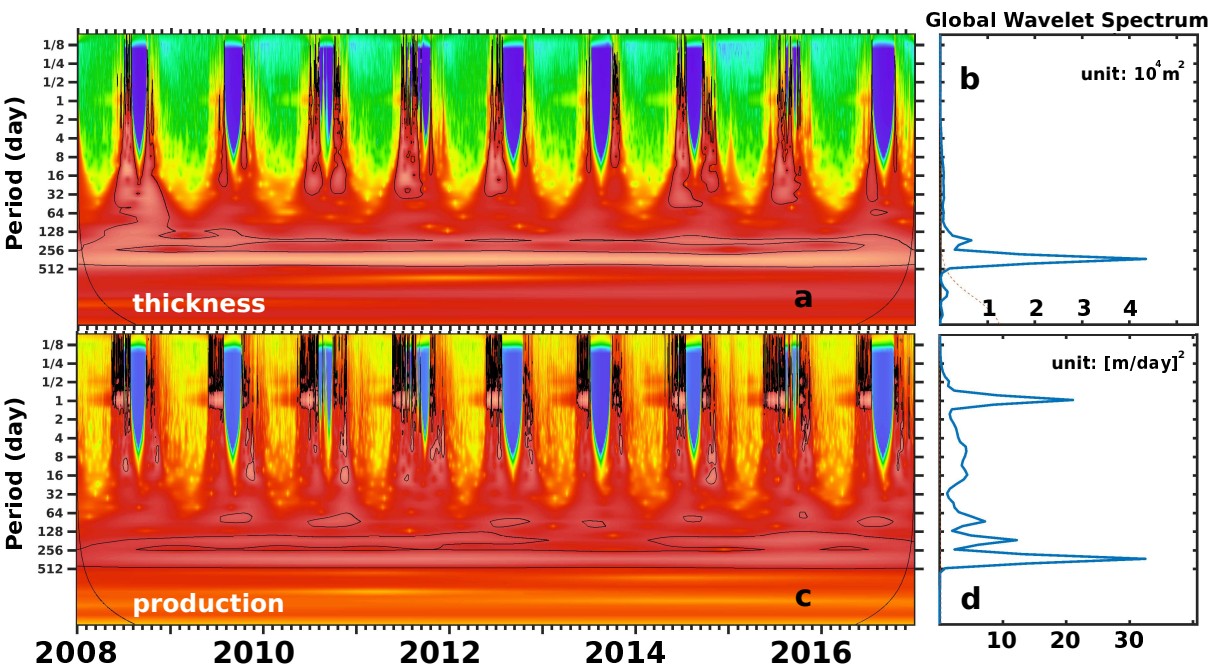

**Figure 8.** Wavelet spectrum (left, black contours highlight the significant oscillations at a confidence level of $95\%$) and global power spectrum (right) of ANHA12 hourly ice thickness (upper panel) and thermodynamic ice production (lower panel) at Cambridge Bay

Figure 8 shows the wavelet spectrum (a and c) and the global wavelet spectrum (b and d) of the ice thickness (a and b) and ice thermodynamic production (c and d) at Cambridge Bay. The thermodynamic ice production was converted to $m\,per\,day$ and normalized based on its standard deviation before performing the wavelet. The seasonal cycle dominates the variability in all the years throughout the analysis period (2008 to 2016), which is also shown in the global wavelet spectrum (Fig. 8b). High

frequency oscillations down to a period of a week are significant before and after the ice free period (Fig. 8a). This supports the notion that the ECCC weekly sampling frequency is good enough during the ice break-up and freeze-up periods. The high frequency ice thickness variation is associated with the diurnal cycle in the thermal ice production (Fig. 8c). The wavelet of the non-thermodynamic ice production has some oscillations at roughly weekly scale, but it is not significant every year in both the break-up and freeze-up periods (not shown). The weekly oscillations in the the dynamic ice production is likely caused by the interaction between the dynamic and thermodynamic processes rather than dynamic process itself. Figure 10 shows an example of the thermal ice production during the 2012 break-up season. The diurnal cycle also exists before the break-up but its amplitude is nearly zero. Thus, it is not significant in the wavelet spectrum (Fig. 8c). Similar results are achieved with the wavelet analysis of the ice fields at Resolute (Fig. 9).

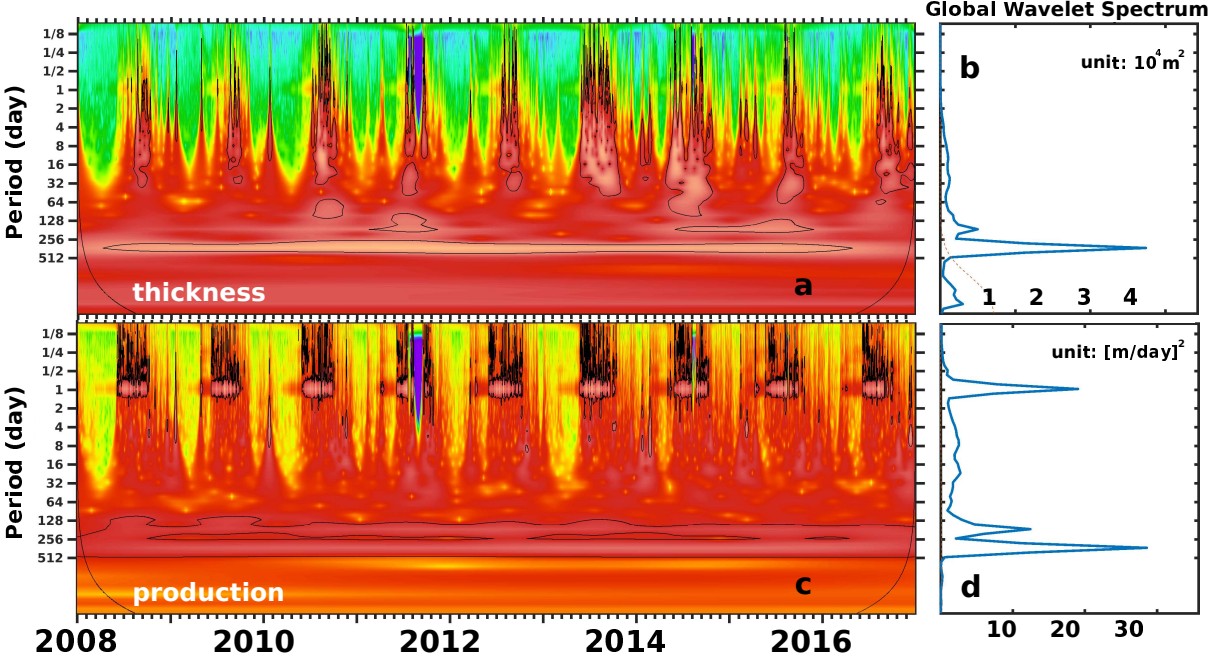

**Figure 9.** Same as Fig. 8 but at Resolute

The thermal ice production diurnal oscillation also shows asymmetric features. For example, in 2012 (Figure 10 and 11), it can reach a melting rate of $0.1 \sim 0.15\,m\,per\,day$ ($0.04 \sim 0.1\,m\,per\,day$) during the day but will fall back to a freezing rate of $0.01 \sim 0.03\,m\,per\,day$ ($< 0.01\,m\,per\,day$) at Cambridge Bay (Resolute). This asymmetric feature is also seen on the season scale, more noticeable during the ice melting period than the freezing period (Figure 8c and 9c).

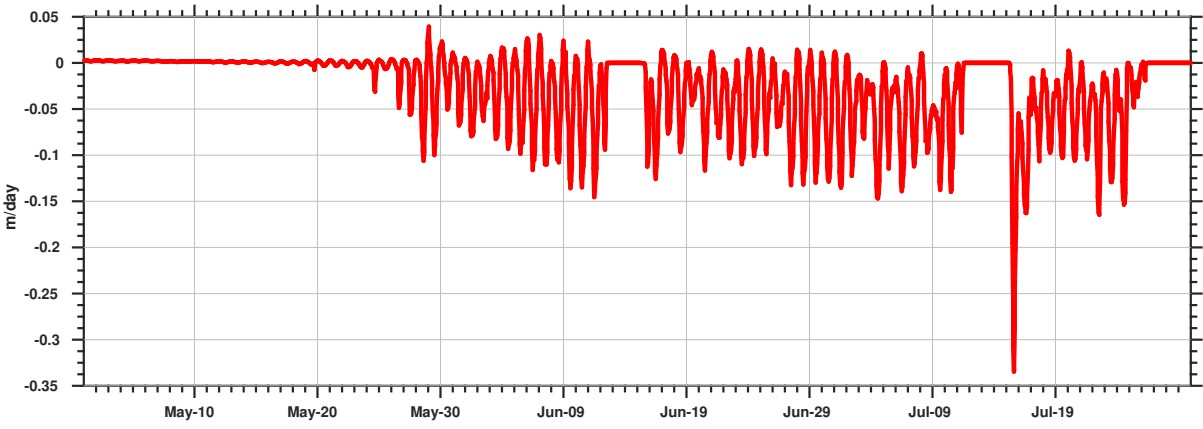

**Figure 10.** Diurnal cycle of thermodynamic ice production (unit: $m/day$) during the break-up season at Cambridge Bay.

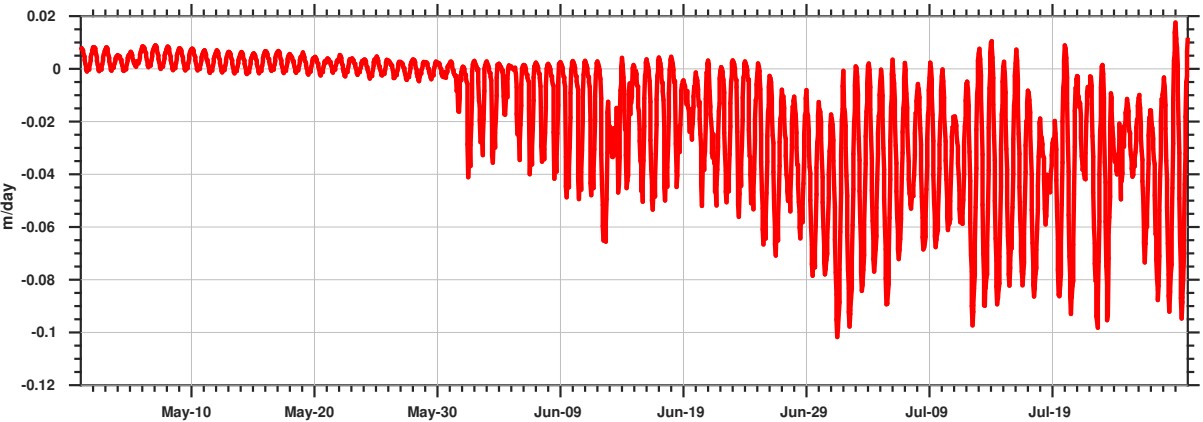

**Figure 11.** Same as Fig. 10 but at Resolute

### 3.3 Ice volume budget

### 3.3.1 Northern CAA

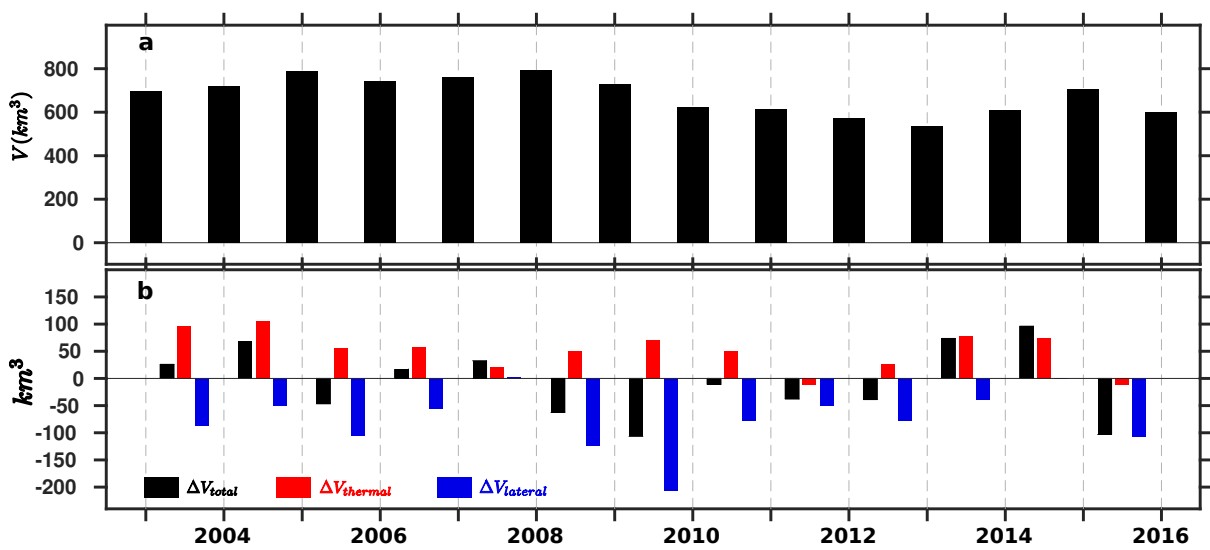

**Figure 12.** Sea ice volume balance in the northern CAA (location see Fig. 1. a) maximum total ice volume (black bars, unit: $km^3$) in each seasonal cycle (September 17 to next September 12). b) the net ice volume change (black bars) between two consecutive years, thermodynamic ice volume change (red bars) and lateral ice volume transport (blue bars) in $km^3$

Fig. 12a shows the maximum total ice volume (referred as "ice volume" hereafter if not mentioned specifically) in the northern CAA (solid black polygon shown in Fig. 1), which is covered by thick ice most of the year. An increase of $14\%$
(from $695\,km^3$ to $789\,km^3$) in the ice volume is shown in the first three years. This is similar to the equilibrium problem we see at Eureka (Fig. 2g). During this period, the thermodynamic growth is the main contributor ($203\,km^3$) while the net lateral ice volume transport ($-138\,km^3$ per year) is out of this region (Fig. 12b), particular through Byam Martin Channel (Fig. 13). The sign convention is defined as positive means transport into the northern CAA regions. The ice volume stabilizes at high values for four years until 2008. After that, a shrinkage of about 1/3 ($792\,km^3$ to $535\,km^3$) in ice volume is simulated over
2008–2013. This reduction is due to large net lateral transport (Fig. 12b), e.g., in 2008 ($-125\,km^3$), 2009 ($207\,km$), and 2012 ($-78\,km^3$). The large lateral transport is not always due to large outflow to the south, e.g., $-102\,km^3$ in 2008 and $-96\,km^3$ in 2010 through Byam Martin Channel and $-48\,km^3$ in 2010 through Penny Strait, but also could be caused by less import (e.g., $8\,km^3$ in 2012) or even export ($-134\,km^3$ in 2009) through the northern gates (Fig. 13). It also shows that large import of ice through the northern gates is usually accompanied by large export to the south, mainly via Byam Martin Channel but
also through Penny Strait in some years, e.g., 2010, 2013 and 2014. Both the thermodynamic and lateral transport (contribution through each major gate as well) experience significant inter-annual variations.

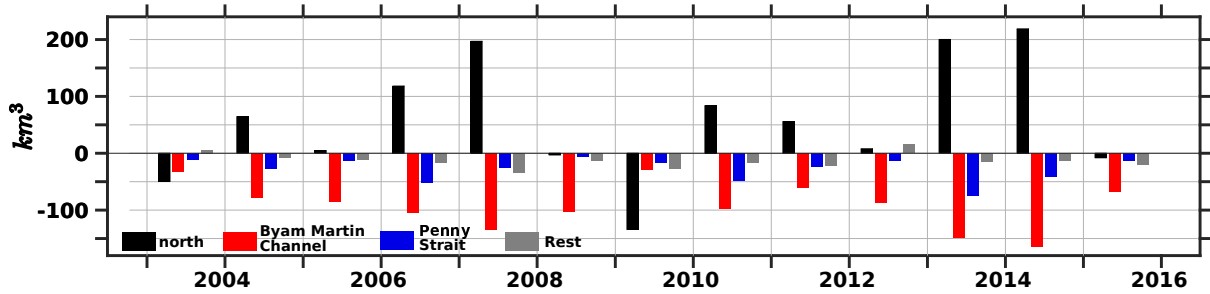

**Figure 13.** Lateral sea ice volume transport (unit: $km^3$, black bars: north gates; red bars: Byam Martin Channel; blue bars: Penny Strait; light gray bars: the rest of the lateral gates) in the northern CAA (location see Fig. 1. a) over the same period defined in Fig. 12.

### 3.3.2 Parry Channel

Parry Channel (dashed black polygon shown in Fig. 1) is the main water channel that connects the Arctic Ocean and Baffin Bay through the CAA (Fig. 1). It starts from M'Clure Strait on the west, running to east by the mouth of Lancaster Sound before entering Baffin Bay. Over the whole simulation, a decrease of $15.2\,km^3$ per year ($r^2 = 0.66$, $p = 0.0004$) in the maximum ice
volume is present (Fig. 14a). Even ignoring the initial increase over the first three years (from $554\,km^3$ in 2003 to $665\,km^3$ in 2005), the downward trend similar ($14.6\,km^3$ per year with $r^2 = 0.58$, $p = 0.0065$). However, this decline is not steady but with inter-annual variability. The minima are found in 2012 ($407\,km^3$) and 2013 ($398\,km^3$), which are more than $20\,\%$ lower than the average, $524\,km^3$. Similar to the northern CAA, thermodynamic growth is the main contributor to the ice volume increase from year to year while net lateral transport functions to deplete the sea ice (Fig. 14b).

Large inflows from M'Clure Strait are simulated in the first two years (Fig. 15), but the direction of sea ice flow can switch from year to year (e.g., $-132\,km^3$ in 2011 and $120\,km^3$ in 2013). The outflow events in 2007 and 2011 are consistent with the ice area flux study in Howell et al. (2013). As significant inter-annual variability in the amount of this ice volume flux is also present, the over all contribution of ice volume into Parry Channel from M'Clure Strait is small, which also supports the finding in Howell et al. (2013).

Major sea ice volume exchanges (Fig. 15) occur at Byam Martin Channel (inflow from the north), M'Clintock Channel (outflow to the south) and Lancaster Sound mouth (outflow to Baffin Bay). On average, annual ice volume fluxes through the first two routes nearly cancel each other ($92\,km^3$ vs $94\,km^3$), which indicates a relatively volume conservation due to high concentrations. The averaged sea ice transport at the east end (via Lancaster Sound mouth) is an export into Baffin Bay ($92\,km^3$ per year), which is closed to an early estimation ($102\,km^3$ per year) from Agnew et al. (2008).

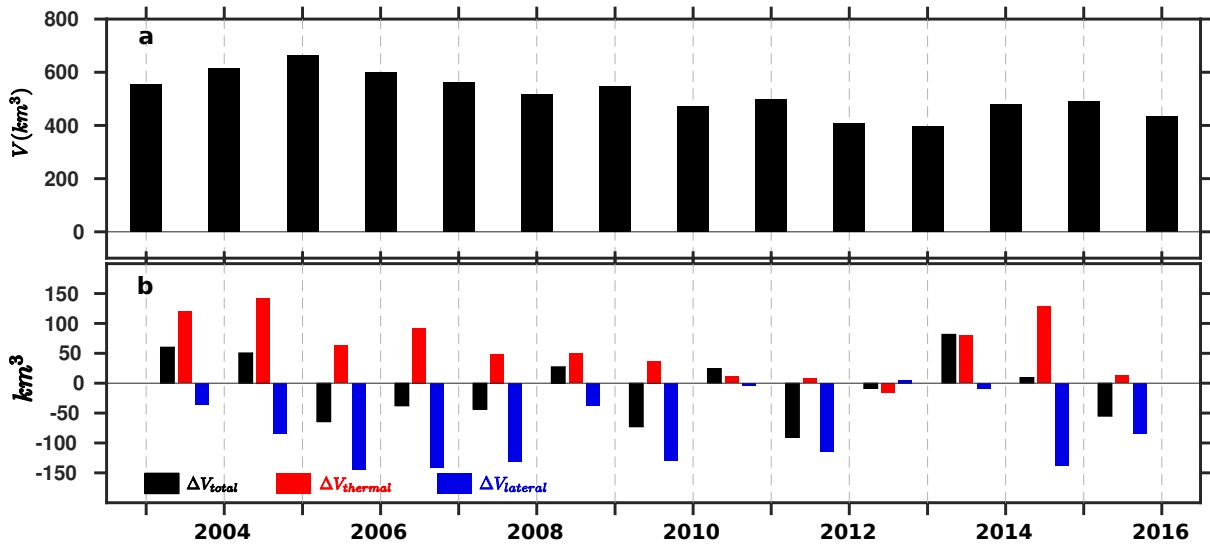

**Figure 14.** Similar to Fig. 12 but within Parry Channel.

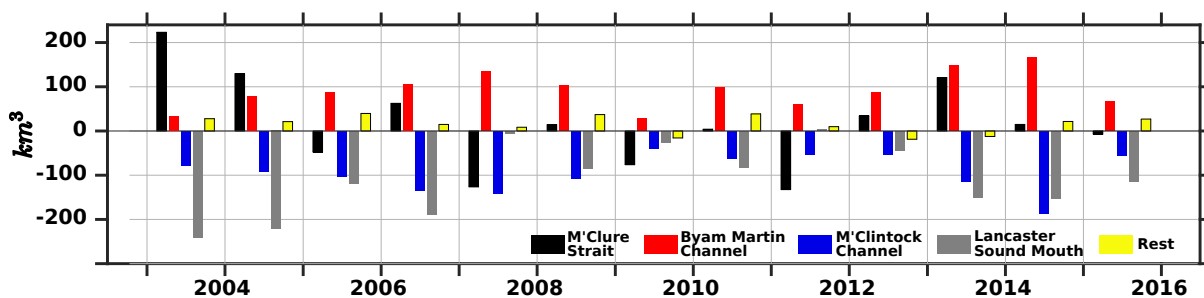

**Figure 15.** Similar to Fig. 13 but for Parry Channel region (black bars: M'Clure Strait; red bars: Byam Martin Channel; blue bars: M'Clintock Channel; light gray bars: Lancaster Sound Mouth; yellow bars: the rest of the lateral gates).

### 3.3.3   Baffin Bay

Baffin Bay (dotted black polygon shown in Fig. 1), bounded by Smith Sound in the north, Jones Sound and Lancaster Sound in the west and Davis Strait in the south, $\sim 672 \times 10^3 \, km^2$, is covered by seasonal sea ice with an averaged maximum ice volume of $895 \, km^3$. No obvious decline is found over the simulation period (Fig. 16a). Although both the local thermodynamic ice growth and lateral ice volume flux show remarkable inter-annual variability (Fig. 16b), the balance between the contributions results in a relatively stable ice volume within the Bay.

Figure 17 shows the lateral ice volume flux is dominated by the inflow from the northern (Smith Sound) and outflow from the south (Davis Strait). On the west side (via Lancaster Sound and Jones Sound), the direction of ice flux is mainly into Baffin Bay, however, the total amount is much smaller than ice volume flux either via Smith Sound or Davis Strait. This is consistent

with other studies (e.g., Tang et al., 2004; Agnew et al., 2008; Sou and Flato, 2009). The averaged export of ice volume flux through Davis Strait is $702\,km^3$ per year with a standard deviation of $147\,km^3$ per year. This number is larger than estimates in Curry et al. (2011) and Curry et al. (2014), $500\,km^3$ and $424\,km^3$ respectively. But they are not very different, taking account of the uncertainties in observations, large inter-annual variability and difference in integration period (Baffin Bay ice volume

maximals are used to determine the integration period in this study). It is more comparable to the $530-800\,km^3$ per year estimated by Kwok (2007). In addition, the low outflow event in 2004 and high outflow event in 2008 agree with Curry et al. (2014). The inflow of ice flux from Smith Sound is $377\,km^3$ per year, which is much larger than the long term mean ($9\,km^3$ per year) in a coarse simulation done by (Sou and Flato, 2009), but closer to their estimate through southern Smith Sound section, i.e., $170\,km^3$. It indicates sea ice in this region is more dynamic in our simulation (Fig. 4). This dynamic feature

is also evidenced in ice motion vector fields derived from enhanced resolution Advanced Microwave Scanning Radiometer (AMSR-E) imagery in Agnew et al. (2008). Relatively large ice fluxes (e.g., $110\,km^3$ per year in 1977-1978 and $136\,km^3$ in 1974-1975) through Smith Sound were also estimated based on satellite images and a mean ice thickness of $2.5\,m$ by Dey (1981). Another way to estimate the ice flux through Smith Sound is based on the ice flux through the north end of Nares Strait (i.e., Robeson Channel). Note ice flux through Smith Sound usually is larger than the sea ice influx through Nares Strait

(Dey, 1981). (Kwok et al., 2010) estimated the annual mean ice volume flux to $141\,km^3$ per year over 2003–2008. The large outflow ($254\,km^3$) event in 2007 through Nares Strait reported by (Kwok et al., 2010) is also seen in our simulation (Fig. 17). Both (Sou and Flato, 2009; Terwisscha van Scheltinga et al., 2010) attributed the much-lower-than-observation ice flux through Nares Strait to wind forcing, which does not have enough resolution to resolve the along-strait winds. With a high resolution wind forcing, Rasmussen et al. (2010) was able to reproduce much reasonable ice flux through this narrow channel.

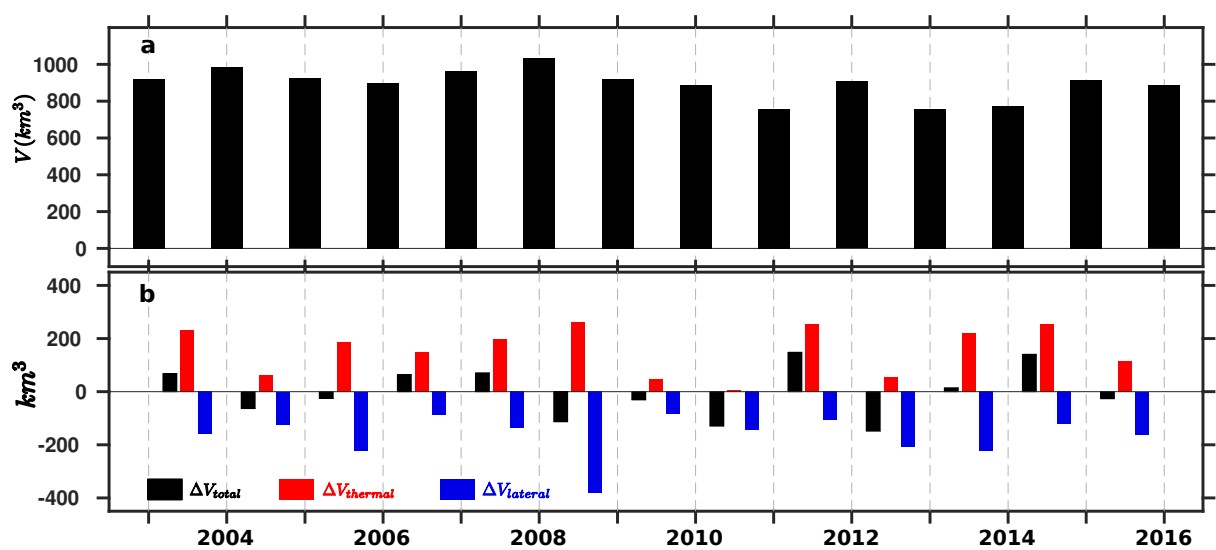

**Figure 16.** Similar to Fig. 12 but within Baffin Bay.

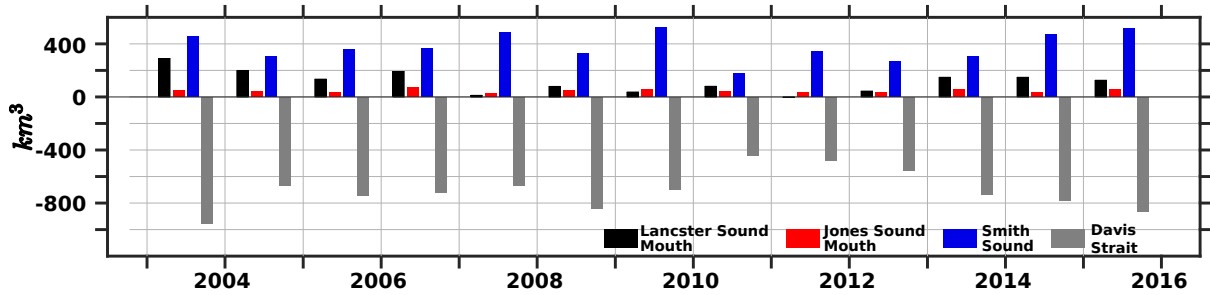

**Figure 17.** Similar to Fig. 13 but for Baffin Bay (black bars: Lancaster Sound Mouth; red bars: Jones Sound Mouth; blue bars: Smith Sound; light gray bars: Davis Strait).

## 4   Summary and discussion

Sea ice thickness is simulated within the CAA with a relatively simple (not multi-category) sea ice model, LIM2, with both $1/4°$ and $1/12°$ resolutions from 2002 to 2016. The model can capture the ice thickness asymmetric seasonal cycle and amplitude. Even with the constraints of the resolution, simulated ice thickness still compares reasonably well with the ECCC
observations at most sites. Increasing model horizontal resolution does not result in much noticeable change/improvement in sea ice thickness simulation, at least between $1/4°$ and $1/12°$. ANHA12 does show differences at Eureka (Fig. 2g), but this is not related to the sea ice model physics but improvements in the local coastline, and thus the regional circulation and the dynamic component (not shown). In general, the difference is not visible. We expect model resolution to play a big role when it resolves much smaller scale, e.g., sufficient to resolve a ridge/lead. This study focuses on the large scale features, e.g., the
simulations can produce reasonable spatial distribution of the thickness (very thick ice in the northern CAA, thick ice in the west-central Parry Channel and thin ice in the eastern and southern regions of CAA). With the help of the numerical model, the ice growth process can be decoupled into thermodynamic and dynamic contributions. Relatively smaller thermodynamic contribution in the winter season is found in the thick ice covered areas, with larger contributions in the thin ice covered regions (which indicates a 1D sea ice model would not be suitable there). We also find the sea thickness can vary quickly (daily to
weekly) during the ice melting and formation seasons, due to the diurnal cycle in the thermodynamic ice production. The thermodynamic ice production is not symmetric for the diurnal cycle and on the seasonal scale (more pronounced during the melting period).

The inter-annual variations of the winter maximum ice volume in the northern CAA, Parry Channel and Baffin Bay are controlled by the thermodynamic growth and lateral transport. While both components demonstrate significant inter-annual
variabilities, there is no clear trend in the winter maximum ice volume within the northern CAA and Baffin Bay regions but a downward trend ($r^2 \approx 0.6$) in Parry Channel region. In the northern CAA, the lateral transport is mainly through the northern gates and Byam Martin Channel but large ice volume flux could also flow south via Penny Strait when there is large inflow through the northern gates. Ice flow via Byam Martin Channel into Parry Channel is balanced by outflow into M'Clintock Channel on average. Eastward sea ice export through Lancaster Sound mouth is a big term in Parry Channel ice volume

budget, but is much smaller than the influx from Smith Sound and outflux through Davis Strait in the ice volume budget of Baffin Bay. These estimates are comparable to limited available studies, however, further evaluations are still in need to confirm the quantities and variations.

Landfast ice exists widely in the CAA (Melling, 2002; Galley et al., 2012; Haas and Howell, 2015; Howell et al., 2016). The sea ice model utilized here does produce zero-motion sea ice (e.g., Fig. 4d), however, more realistic physical parameterizations (e.g., Lemieux et al., 2016) are not applied in our simulations yet. Improvements are expected in both the sea ice thickness and dynamics when including such parameterizations in the future.

Another ice model physics related process is the snow depth at the surface. Brown and Cote (1992), Dumas et al. (2006) and Howell et al. (2016) pointed out that the snow thickness plays a major role in a sea ice thickness simulation. The snowfall data from CORE-II has a monthly resolution, which is possibly too coarse temporally. This leads the snow depth to drop to close to zero quickly during the the first year of the simulation. This never happens with the hourly CGRF forcing, as well as other datasets with daily snowfall (Hayashida, 2017).

A large source of bias is related to the radiation fluxes (shortwave and longwave) at the atmosphere-ocean interface. The CGRF product use the GEWEX correction (more details in Smith et al. (2014)) to minimize the bias in the original output from the atmospheric model. However, it was found that this correction is not needed for years after 2011 (Paquin, 2017). A test run was done to estimate the influence of this correction. A large impact was seen in the Arctic Ocean but not over most of the CAA (not shown) except the north. Without the GEWEX correction, the model tends to simulate thinner ice with a thinner snow depth at the same time, e.g., at Eureka (Fig. 18). The impact is smaller at Alert and Alert LT1 and negligible for the rest of the ECCC sites used in this study. Further investigation is undergoing.

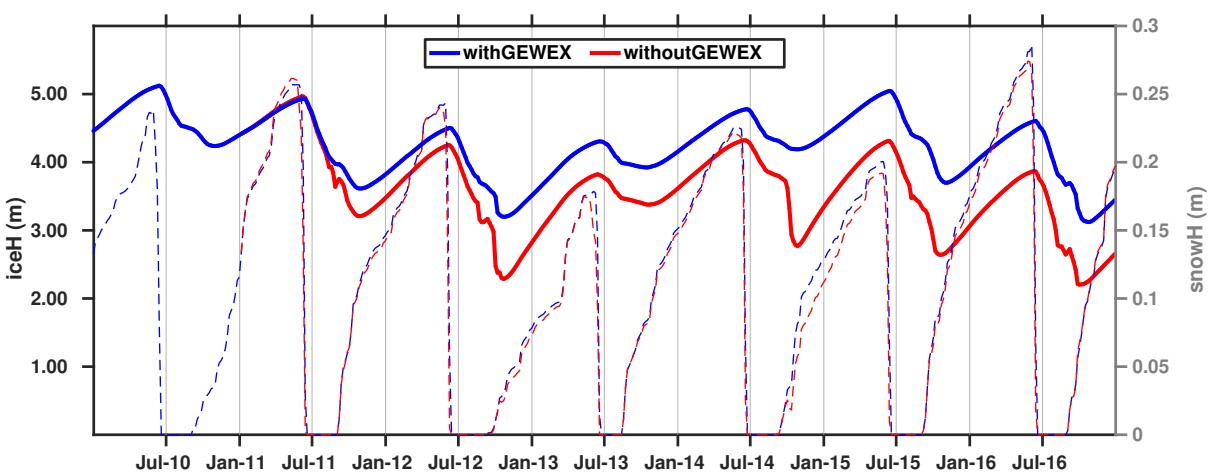

**Figure 18.** Simulated ice thickness (solid lines, left y-axis, unit: meters) and snow depth (dashed lines, right y-axis, unit: meters) at Eureka in a ANHA4-CGRF simulation with GEWEX (blue) and without GEWEX (red) radiation flux correction

Whitefield et al. (2015) showed the heat flux carried by rivers from land into the ocean plays a role in regional sea ice simulations. Here the river water temperature is assumed to same as the simulated sea surface temperature in the grid cell

where the runoff is distributed in the model. Also the salinity of the river water is assumed to be fresh, i.e., its salinity is set to zero. Another factor that may affect the local ice thickness is the tides (Luneva et al., 2015), which is not included in the simulations used in this study as well. All these factors will be considered in the future studies.

*Data availability.* For access to the model data contact P.G. Myers (pmyers@ualberta.ca)

*Competing interests.* NONE

*Acknowledgements.* For access to the model data contact P.G. Myers (pmyers@ualberta.ca). We gratefully acknowledge the financial and logistic support of grants from the Natural Sciences and Engineering Research Council (NSERC) of Canada. These include a Discovery Grant
(rgpin 227438-09) awarded to P.G.M., Climate Change and Atmospheric Research Grants (VITALS - RGPCC 433898 and the Canadian Arctic Geotraces program - RGPCC 433848), and Polar Knowledge (432295). We are grateful to the NEMO development team and the Drakkar project for providing the model and continuous guidance, and to Westgrid and Compute Canada for computational resources. We also thank G. Smith for the CGRF forcing fields that made available by Environment and Climate Change Canada. We appreciate the Environment and Climate Change Canada New Arctic Ice Thickness Program for providing valuable sea ice thickness measurements used
in this study. Greenland freshwater flux data analyzed in this study is that presented in Bamber et al. (2012) and is available on request as a gridded product. We thank NCAR/UCAR for making Dai and Trenberth Global River Flow and Continental Discharge Dataset availabile. We acknowledge WCRP/CLIVAR Ocean Model Development Panel (OMDP) for sponsoring and organizing the Coordinated Ocean-sea ice Reference Experiments dataset (CORE). We also acknowledge Mercator Ocean for providing the GLORYS model output for initial and open boundary conditions. The GLORYS reanalysis project is carried out in the framework the European Copernicus Marine Environment
Monitoring Service (CMEMS).

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
