# Peer review of "Thermodynamic and dynamic ice thickness contributions in the Canadian Arctic Archipelago in NEMO-LIM2 numerical simulations"

_The Cryosphere, 2017_

## Referee Comment (RC1) · Anonymous Referee #1 · 4 Nov 2017

Thermodynamic and Dynamic Ice Thickness Changes in the Canadian Arctic Archipelago in NEMO-LIM2 Numerical Simulations

Hu and others developed and presented a high resolution model to look at simulated sea ice thickness in the CAA from 2002-2016. They compared their model output to ice thickness from the Canadian Ice Service. From the title, I was expecting the authors to look at thermodynamic and dynamic processes contributing to variability and change but they only scratched the surface. Overall, I feel the authors did not really use the model to its full potential. I think some additional analysis and interpretation is required and I offer the following suggestions:

[Figure]

1. I was very surprised the authors chose not to look at the entire CAA ice thickness time series separating out dynamic and thermodynamic from 2002-2016 similar to their Figure 4 site specific plots. Doing this would probably illustrate when (and then where) changes in these processes are occurring over the longer term record and this certainly would allow for more discussion. Also, related to thickness distribution changes they could investigate if the model can identify the regions of very thin ice (i.e. invisible polynya's) similar to Melling et al. 2015 and look at longer term variability.

2. Section 3.2.1 provides useful information by separating the dynamic and thermodynamic component of ice thickness but it would also be better use the model to identify locally grown MYI from MYI advected from the Arctic Ocean. It would be also be useful to spatially illustrate changes in the source of MYI. Furthermore, why not construct an ice mass budget for the CAA and look at how it changes from 2002-2016? These additions would give more substance to the manuscript.

3. Why did the author's chose to start the study in 2002? I would think looking at longer term changes would provide more useful information to the readers and provide more opportunity to compare to Sou and Flato (2009).

Specific comments: Page 1, Line 1: "Sea ice thickness evolution within the Canadian Arctic Archipelago (CAA) is of great interest." Why?

Page 2, Line 5 National security issues?

Page 2, Line 5 Replace "opening" with "using"

Page 2, Line 13 The dates of that study are from 1979-2008. Does this statement still hold true? Looking at recent work posted on The Cryosphere Discussion appears to indicate large changes have occurred in the last 10-years which could negate that statement. Rephrase. See Mudryk et al. (2017) in The Cryosphere Discussions.

Page 2, Line 23 Replace "export" with "transport" and replace "in the past" with "known to occur."

[Figure]

Page 6, Lines 9 to 18 I think it is important to state that the sites are all on landfast sea ice.

Page 6, Table 2 I think the reader could better identify with actual place names and not an acronym.

Page 8, Line 10 It would be useful to include some correlation coefficient values for comparison.

Page 24 Could you confirm this by looking at a long time series? I'm puzzled by the 2002 start date.

Page 11, Line 5 It is been shown MYI flows down into these regions. How much is convergence compared to thick MYI?

Page 12, Section 3.3 A lot of methodology and techniques are being introduced the results section. Suggest moving to methods.

Figure 2. There are no y and x-axis labels.

Figures 4-6 No y-axis labels. Also, why not produce this figure for the entire CAA? That would be more useful and also show changes in ice thickness (dynamic versus thermodynamic) over a 15-year period. I don't feel the work reflections the title. See major suggestion 1.

References:

Mudryk, L., Derksen, C., Howell, S., Laliberté, F., Thackeray, C., Sospedra-Alfonso, R., Vionnet, V., Kushner, P., and Brown, R.: Canadian Snow and Sea Ice: Trends (1981–2015) and Projections (2020–2050), The Cryosphere Discuss., https://doi.org/10.5194/tc-2017-198, in review, 2017.

Melling, H., C. Haas, and E. Brossier (2015), Invisible polynyas: Modulation of fast ice thickness by ocean heat flux on the Canadian polar shelf, J. Geophys. Res. Oceans, 120, 777–795, doi:10.1002/2014JC010404.

---

## Referee Comment (RC2) · Anonymous Referee #2 · 13 Nov 2017

Manuscript: Thermodynamic and Dynamic Ice Thickness Changes in the Canadian Archipelago in NEMO-LIM2 Numerical Simulations

General Review:

This manuscript compares simulated sea-ice thickness profiles with those from observation from a handful of locations in the Canadian Arctic Archipelago. I have a hard time pinpointing the overall purpose of the manuscript. The observations seems to indicate largely thermodynamic growth and melt i.e. relatively smooth seasonal cycles with production of 1.5 - 2m of ice during the winter which then melts out during the summer. The model simulations seem to capture this well at some locations, while at

other locations there are discrepancies between the obs. and the simulations. This point is discussed in passing in the manuscript; however a much more thorough explanation of this issue is of interest. This is particularly problematic considering that Dumas et al (2006) have shown that a 1-D thermodynamic model can largely recreate observed sea-ice thickness in the CAA at the same locations.

Furthermore, I find the methods used to compare the in-situ (point) observations to the model output to be unsatisfactory. The sea-ice thickness from the simulations as described by the author is a grid-cell mean - i.e. already smoothed compared to the observations. Yet, the authors go on to further smooth the model output with a 9 grid cell stencil. This issue likely does not substantially change the results of the comparison since the ice growth/melt is largely thermodynamic (relatively smooth and large decorrelation length scale). However, this choice to further smooth the fields is confusing and does not make logical sense.

The authors separate the dynamic and thermodynamic contributions to the change in sea-ice thickness during the Arctic winter. This is the most interesting part of the manuscript and should be expanded on. The results indicate a net thinning of the ice in Baffin bay due to dynamics and an associated thermodynamic growth. I suspect this is due to the formation of polynyas and the resulting first year ice production in Nares Strait region. It would be interesting if the authors could show timeseries of the separated thermodynamic and dynamic growth at some of these points to see if/when the polynya forms every year. It would also be useful to add a panel to figure 3 which shows the difference between panel a and panel b in order to see the full \Delta h field.

In my opinion, one of the interesting questions that arises from this manuscript is: why does the model produce much thicker ice at Eureka and Alert compared to the observations? Why is the magnitude of the interannual variability at Alert in some simulations/years so much greater than all other locations. The manuscript would be more relevant and useful to the community if this was investigated and a solution proposed to fix this issue.

The manuscript goes on to present a complicated wavelet analysis to study the seasonal and diurnal cycle in sea-ice thickness. This analysis is entirely unnecessary as it is expected (and obvious) that there is a seasonal cycle in sea ice thickness. Furthermore, the seasonal cycle is already clearly shown in the obs and simulations in figure 2. The analysis of the diurnal cycle dummer is also unnecessary as the conclusion is exactly what must be the case- i.e. daytime melt overwhelming slight nighttime freezing.

I would also encourage the authors to generally use a simpler and more concise sentence structure. There are many long and confusing sentences which makes it difficult to follow the authors' arguments.

Particular issues:

P1 L12-13: It is well known that thermodynamic growth of ice is inversely proportional to thickness. This is not a contribution of your work.

P2 L8: 10% of the sea-ice area? Or volume? Please clarify

P2 L20: Landfast Ice implies u=v=0, not just 100% concentration

P2 L28: Model simulations are not substitutes for in-situ observations. Your manuscript is showing discrepancies between the obs and the simulations!

P4: L17-20: There are many more recent and informative studies regarding the time scale and decay of the artificial elastic waves. From my experience, your choices of number of subcycles is far too low particularly at 1/12 degree resolution. See for instance: Lemieux et al (2012), Boullion et al (2013), Kimmritz et al (2016), Williams et al (2017).

P4 L27: How are the 33km wind fields used to force the simulations with different spatial resolutions? There seems to be many issues which could arise here.

P5 L5: It is unclear how the CORE II simulations incorporate the inter-annual variablity

of the atmospheric forcing. What is the climatological mean? This deserves more explanation.

P7 - see 3rd paragraph in general review

P8 - see 1st and 4th paragraph in general review

P8 L25 It's not clear how the data assimilation is taking place. What fields are being assimilated, and in which simulations? How does this affect the results? What if no assimilation is done?

P9 Fig 2: All of the observed timeseries look similar in this figure. Perhaps another figure showing the differences due to location would be useful.

P12-15: Full timeseries of these fields would be much more interesting to see at these locations rather than seasonal cycles. This would allow us to see if there is a correlation between particular dynamic events and the thermodynamics feedbacks that we expect. Perhaps keep the seasonal cycles as well for completeness.

P16-20: I do not see what this analysis adds to the story. We already see that there is a seasonal cycles and it must be that daytime melt outweighs nighttime freezing during the melt season.

Some grammar / technical issues:

P1 L11: A relatively small

P1 L22- P2 L4: Confusing, rephrase

P2 L13: Rephrase. Also remove the quotes around statistically significant

P2 L17: There are

P2 L24: conditions

P4 L19: "can" does not make sense here. No-slip boundary conditions define that the velocity is zero at the coast line

[Figure]

P6 L6: This sentence is unclear and further explanation of the assimilation process is required.

P6 L12: delete "only"

P6 L13: delete "period"

P7: L9: calculation

P9 L3: "Cambridge Bay" rather than "the Cambridge Bay"

---

## Referee Comment (RC3) · Anonymous Referee #3 · 28 Nov 2017

Thermodynamic and Dynamic Ice Thickness Changes in the Canadian Arctic Archipelago in NEMO-LIM2 Numerical Simulations

General comments:

This is an interesting study, comparing sea ice thickness simulations from a numerical model with landfast ice thickness observations at eight sites in the Canadian Arctic Archipelago, separating simulated changes in ice thickness into thermodynamic and dynamic contributions, and describing diurnal oscillations in ice thickness and thermal ice production. However, I feel that the purpose of the work is not clearly articulated. I suggest it could say something like "first, to evaluate the skill of a numerical model in

simulating sea ice thickness by comparing the simulations with observations of landfast ice thickness at several sites in the CAA. Two features of the simulations will be then be discussed: 1) the relative importance . . .".

I also feel that the paper does not make sufficiently clear the difference in the properties of the observation data versus the simulation data. The observation data represents immobile level first-year (seasonal) ice of uniform thickness that forms close to shore, and is forced by thermodynamic processes. The simulation data (page 8, line 12) generally represents ice found beyond the near-shore ice and is a mixture of deformed (ridged/rafted) and level first-year ice, young ice and old (perennial) ice, is mobile for part of the year, and is forced by both thermodynamic and dynamic processes. The degree to which we should expect them to agree therefore depends on the concentration of old ice and deformed ice, differences in the timing of freezeup/breakup, etc.

I think that more detail is required to describe the skill of the model. The summary (but not the abstract) mentions the capability of capturing the seasonal cycle and amplitude of ice thickness. This would be clearer if the seasonal cycles were plotted as in Howell et al. (2016). In addition, such a plot would more clearly show the differences/agreement between model results and observations at Resolute and Cambridge Bay. Perhaps the dynamic processes in Figures 4 and 5 could then be used to explain, in part, these differences. Does the model have any significant skill with respect to interannual variability (or does it not, because of snow depth variations on small horizontal scales)?

Minor comments:

Page 1, lines 3-6: "the model captures well the general spatial distribution . . . ($\sim$4 m and thicker)". While this may be true, the model was compared with landfast ice thickness observations (first year ice only, no old ice or deformed ice), that are generally not much greater than 2 m. Why not describe a general comparison with published data from IceSat, CryoSat or other sources (e.g. Laxon et al., 2013; Tilling et al.,

2015), which include the thicker ice types?

Page 1, lines 6-8: What is meant by "compares well"? Do you mean the seasonal cycles and amplitudes, as stated in the summary? Is agreement with first-year landfast ice better in the south because there are low concentrations of old ice?

Page 1, line 13: Add "at two sites" after "ice fields"

Page 2, line 34: "this downward trend is mostly associated with changes in snow depth". The meaning of this is not clear. Do you mean that in most cases, the downward trend in ice thickness is associated with a positive trend in snow depth (since ice thickness is negatively correlated with snow depth)? Only one of the cases had a significant trend in snow depth, and it was negative, not positive.

Page 3, line 3-4: Change "a sea ice model" to "several sea ice models"?

Page 6, line 12: Were three of the 11 stations omitted from the analysis because they were on lakes?

Page 6, line 16 and elsewhere: The paper would be much easier to read if the full names (not acronyms) were used for the station locations.

Page 8, line 10: The 3 sites with poor agreement between simulations and observations are in areas with significant concentrations of old ice, while the sites with reasonable agreement are in areas without (see Canadian Ice Service (2011)). Is this the basic reason for the poor agreement at the 3 sites?

Page 8, line 11: I suggest adding a plot of the seasonal cycles of the models and observations (as in Howell et al 2016, Figure 8). This would make it easier to visualize the asymmetric seasonal cycles and summarize the differences in amplitude etc. between the various models.

Page 8, line 21: "too thick sea ice". What would be a realistic sea ice thickness, based on the literature, given that there are significant concentrations of old ice in the area?

Page 10, line 6-7: The meaning isn't clear. "Thus, it is likely due to another physical process such as advection from surrounding areas" (?)

Page 17: I suggest reversing the order of Figure 8 and 9, so that they are in the same order as in the text.

Technical comments:

Page 1, Line 21: "overturning"

Page 2, line 17: "there are still"

Page 2, line 30: "evaluated the"

Page 4; Table 1: "subcycling" (?)

Page 6, lines 16 and Table 2: Change "Carol" to "Coral".

Page 8, line 19: Add "(Fig. 2c and d)".

Page 8, line 20: Change "MEU" to "WEU".

Page 8, line 24: "green line" (add space)

Page 8, line 34: Add ".".

Page 10, line 21: "just south of the site YRB" (?)

Page 12, line 4: "spatial"

Page 16, line 9: "supports the notion that" (?)

Page 19, line 4: "constraints"

References

Canadian Ice Service, 2011. Sea Ice Climatic Atlas: Northern Canadian Waters (http://publications.gc.ca/site/eng/441147/publication.html )

[Figure]

Howell, S. E. L., Laliberté, F., Kwok, R., Derksen, C., and King, J.: Landfast ice thickness in the Canadian Arctic Archipelago from observations and models, The Cryosphere, 10, 1463-1475, https://doi.org/10.5194/tc-10-1463-2016, 2016.

Laxon S., W., K. A. Giles, A. L. Ridout, D. J. Wingham, R. Willatt, R. Cullen, R. Kwok, A. Schweiger, J. Zhang, C. Haas, S. Hendricks, R. Krishfield, N. Kurtz, S. Farrell and M. Davidson (2013), CryoSat-2 estimates of Arctic sea ice thickness and volume, Geophys. Res. Lett., 40, 732–737, doi:10.1002/grl.50193.

Tilling, R. L., Ridout, A., Shepherd, A., and Wingham, D. J.: Increased Arctic sea ice volume after anomalously low melting in 2013, Nat. Geosci., 8, 643–646, 2015.
* * *

---

## Author Comment (AC1) · 17 Jan 2018

article graphicx float booktabs [justification=centering]caption

[Figure]

**Thermodynamic and Dynamic Ice Thickness Changes in the Canadian Arctic Archipelago in NEMO-LIM2 Numerical Simulations**

Xianmin Hu, Jingfan Sun, Ting On Chan, and Paul G. Myers

January 17, 2018

**Reply to Reviewer 1:**
We thank reviewer 1 for pointing out the misleading title, as well as their other comments. We have updated the draft. Details are given as follows.

**Answer to general comments:** "Hu and others developed and presented a high resolution model to look at simulated sea ice thickness in the CAA from 2002-2016. They compared their model output to ice thickness from the Canadian Ice Service. From the title, I was expecting the authors to look at thermodynamic and dynamic processes contributing to variability and change but they only scratched the surface. Overall, I

feel the authors did not really use the model to its full potential. I think some additional analysis and interpretation is required and I offer the following suggestions:

1. I was very surprised the authors chose not to look at the entire CAA ice thickness time series separating out dynamic and thermodynamic from 2002-2016 similar to their Figure 4 site specific plots. Doing this would probably illustrate when (and then where) changes in these processes are occurring over the longer term record and this certainly would allow for more discussion. Also, related to thickness distribution changes they could investigate if the model can identify the regions of very thin ice (i.e. invisible polynya's) similar to Melling et al. 2015 and look at longer term variability."

**We apologize for the old misleading title. We were not focusing on the inter-annual variability but wanted to present the spatial distribution and seasonal cycle in the original draft. "Change" was used in the title to avoid "negative seaice growth" (melting period). However, it turned out be more misleading. We changed the title to "Thermodynamic and dynamic ice thickness contributions in the Canadian Arctic Archipelago in NEMO-LIM2 numerical simulations" to make it clear. In addition, we added a new chapter on ice volume budget in the northern CAA, Parry Channel and Baffin Bay to better describe the ice changes in our study area. We do see some polynya features, e.g., relatively thinner thickness, in the ice thickness field or the dynamic component of the ice thickness contribution, but the simulations does not produce thin enough or a good enough concentration field to investigate the polynya processes in detail. We keep that in mind for future investigation.**

"2. Section 3.2.1 provides useful information by separating the dynamic and thermodynamic component of ice thickness but it would also be better use the model to identify locally grown MYI from MYI advected from the Arctic Ocean. It would be also be useful to spatially illustrate changes in the source of MYI. Furthermore, why not construct an ice mass budget for the CAA and look at how it changes from 2002-2016? These

additions would give more substance to the manuscript."

**A section about ice volume budget is added to provide some related information. Note that with LIM2, we can not identify the MYI directly.**

"3. Why did the author's chose to start the study in 2002? I would think looking at longer term changes would provide more useful information to the readers and provide more opportunity to compare to Sou and Flato (2009)."

**This is due to the availability of the atmospheric forcing. The high resolution CGRF data set goes back to 2002 only. Even with other forcing data, the computation cost, limits our ability to carry out long runs of 1/12 degree resolution.**

**Answer to specific comments:**

- "Page 1, Line 1: Sea ice thickness evolution within the Canadian Arctic Archipelago (CAA) is of great interest." Why?"
  **Changed to "Sea ice thickness evolution within the Canadian Arctic Archipelago (CAA) is of great interest to science, as well as local communities and their economy"**

- "Page 2, Line 5 National security issues?"
  **Changed "secucrity" to "safety"**

- "Page 2, Line 5 Replace "opening" with "using""
  **Done as suggested.**

- "Page 2, Line 13 The dates of that study are from 1979-2008. Does this statement still hold true? Looking at recent work posted on The Cryosphere Discussion appears to indicate large changes have occurred in the last 10-years which could

negate that statement. Rephrase. See Mudryk et al. (2017) in The Cryosphere Discussions."

**Updated based on the recommended reference. The revised version is " Reduction in the September MYI cover is also found to be -6.4% per decade until 2008 (Howell et al., 2009). But this trend was not "yet statistically significant" due to the inflow of MYI from the Arctic Ocean mainly via the Queen Elizabeth Islands (QEI) gates in August to September (Howell et al., 2009). With extended data in recent years (until 2016), Mudryk et al. (2017) showed that the summer MYI decline rate has almost doubled"**

- "Page 2, Line 23 Replace "export" with "transport" and replace "in the past" with "known to occur.""
**Changed as suggested.**

- "Page 6, Lines 9 to 18 I think it is important to state that the sites are all on land-fast sea ice."
**Added text to make is clear the observations are landfast ice. To evaluate the performance of the model in terms of ice thickness, simulated ice thickness is compared to the observed landfast ice data from Environment and Climate Change Canada (ECCC) New Icethickness Program (hereafter ECCC thickness).**

- "Page 6, Table 2 I think the reader could better identify with actual place names and not an acronym."
**Agree. We use full names in both table 2 and texts now in the revised version. Acronyms are now used only in figure 1 to keep it concise.**

- "Page 8, Line 10 It would be useful to include some correlation coefficient values for comparison."
**We think the correlation in our case could be biased by the seasonal cycle. In the revised version, we added the seasonal cycle plot. It provides more**

**information to our current comparison. Our time series might be too short
to provide a robust interannul correlation.**

- "Page 24 Could you confirm this by looking at a long time series? I'm puzzled by
the 2002 start date."
**It is due to the availability of our high resolution atmospheric forcing data.**

- "Page 11, Line 5 It is been shown MYI flows down into these regions. How much
is convergence compared to thick MYI?"
**In LIM2, we can not identify the MYI. Thus, we can not accurately do this
task. We hope the reviewer is happy with our new ice budget section.**

- "Page 12, Section 3.3 A lot of methodology and techniques are being introduced
the results section. Suggest moving to methods."
**Moved the the methodology and techniques texts to method section, "2.3
Wavelet analysis" as suggested.**

- "Figure 2. There are no y and x-axis labels."
**Added as suggested.**

- "Figures 4-6 No y-axis labels. Also, why not produce this figure for the entire
CAA? That would be more useful and also show changes in ice thickness (dy-
namic versus thermodynamic) over a 15-year period. I don't feel the work reflec-
tions the title. See major suggestion 1."
**Added the y-axis labels. We were focusing on the spatial distribution over
the CAA rather than how it behaves over the entire region. It does show
significant spatial variability. It is better to average over a region with fields
that do not vary a lot in space. Anyway, we think the new ice budget section
provides interesting information.**

---

## Author Comment (AC2) · 17 Jan 2018

article graphicx float booktabs [justification=centering]caption

[Figure]

**Thermodynamic and Dynamic Ice Thickness Changes in the Canadian Arctic Archipelago in NEMO-LIM2 Numerical Simulations**

Xianmin Hu,Jingfan Sun, Ting On Chan and Paul G. Myers

January 17, 2018

**Reply to Reviewer 2:**
**We thank reviewer 2 for pointing out various issues with our manuscript. Here are our responses.**

**Answer to general comments:**
"This manuscript compares simulated sea-ice thickness profiles with those from observation from a handful of locations in the Canadian Arctic Archipelago. I have a hard

time pinpointing the overall purpose of the manuscript. The observations seems to indicate largely thermodynamic growth and melt i.e. relatively smooth seasonal cycles with production of 1.5 - 2m of ice during the winter which then melts out during the summer. The model simulations seem to capture this well at some locations, while at other locations there are discrepancies between the obs. and the simulations. This point is discussed in passing in the manuscript; however a much more thorough explanation of this issue is of interest. This is particularly problematic considering that Dumas et al (2006) have shown that a 1-D thermodynamic model can largely recreate observed sea-ice thickness in the CAA at the same locations.

Furthermore, I find the methods used to compare the in-situ (point) observations to the model output to be unsatisfactory. The sea-ice thickness from the simulations as described by the author is a grid-cell mean - i.e. already smoothed compared to the observations. Yet, the authors go on to further smooth the model output with a 9 grid cell stencil. This issue likely does not substantially change the results of the comparison since the ice growth/melt is largely thermodynamic (relatively smooth and large decorrelation length scale). However, this choice to further smooth the fields is confusing and does not make logical sense.

The authors separate the dynamic and thermodynamic contributions to the change in sea-ice thickness during the Arctic winter. This is the most interesting part of the manuscript and should be expanded on. The results indicate a net thinning of the ice in Baffin bay due to dynamics and an associated thermodynamic growth. I suspect this is due to the formation of polynyas and the resulting first year ice production in Nares Strait region. It would be interesting if the authors could show timeseries of the separated thermodynamic and dynamic growth at some of these points to see if/when the polynya forms every year. It would also be useful to add a panel to figure 3 which shows the difference between panel a and panel b in order to see the full $\Delta_h$ field.

In my opinion, one of the interesting questions that arises from this manuscript is: why does the model produce much thicker ice at Eureka and Alert compared to the

observations? Why is the magnitude of the interannual variability at Alert in some simulations/years so much greater than all other locations. The manuscript would be more relevant and useful to the community if this was investigated and a solution proposed to fix this issue.

The manuscript goes on to present a complicated wavelet analysis to study the seasonal and diurnal cycle in sea-ice thickness. This analysis is entirely unnecessary as it is expected (and obvious) that there is a seasonal cycle in sea ice thickness. Furthermore, the seasonal cycle is already clearly shown in the obs and simulations in figure 2. The analysis of the diurnal cycle dummer is also unnecessary as the conclusion is exactly what must be the case- i.e. daytime melt overwhelming slight nighttime freezing.

I would also encourage the authors to generally use a simpler and more concise sentence structure. There are many long and confusing sentences which makes it difficult to follow the authors' arguments."

**First, in our original comparison, we did not address clearly the differences between in-situ observed and simulated ice thickness. This is pointed out by #3 reviewer. The observation (ECCC site data used in this study) represents the "immobile level first-year (seasonal) ice of the uniform thickness that forms close to shore, and is forced by thermodynamic processes". Second, we have to consider the differences between 1d and 3d simulations. The on site ice thickness can be better or more easily captured by the 1d simulation, e.g., in Dumas et al. (2006). But, in 3d coupled ocean and sea ice simulations, it is very difficult to reproduce such local behavior because of the resolution of both the model and atmospheric forcing data. However, we need 3d simulations to better understand seaice processes, particularly when they are not dominated by thermodynamics and their spatial distribution.**

An estimate of the skill of the model is needed but very limited time series are available for a fair comparison. Neither the interpolation or the nearest point method is perfect in such comparisons because it is essentially not resolved by such simulations. Thus, we do not think the method used in this study itself affects our results here.

The differences between the observed and simulated ice thickness also explain the reviewer's question on ice thickness at Eureka and Alert.

The polynya related questions are great and interesting questions. We think they could be further investigated in a future study. In this study, we focus more on the big picture aspects of the simulations.

For the wavelet analysis, the simulations do show the fact (seasonal and diurnal cycle) that we might have expected from the real world. Instead of thinking "it must be the case", we prefer to show and quantify it. In addition, we do think it is a good thing to see that the model can reproduce the basic physical processes because models do not always do the right thing. Thus, we still think we do have some scientific contribution in this study.

**Answer to particular issues:**

- "P1 L12-13: It is well known that thermodynamic growth of ice is inversely proportional to thickness. This is not a contribution of your work."
  **We think this sentence should be read considering the contex of the whole abstract. We are trying to describe what we see based our analysis (thus it is part of our results), but not to declare that we are the first one who found "thermodynamic growth of ice is inversely proportional to thickness".**

- "P2 L8: 10% of the sea-ice area? Or volume? Please clarify"

**Added "volume" in the text.**

- "P2 L20: Landfast Ice implies u=v=0, not just 100% concentration"
  **Added "without motion" to the text.**

- "P2 L28: Model simulations are not substitutes for in-situ observations. Your manuscript is showing discrepancies between the obs and the simulations!"
  **We did not attempt to express that model simulations can replace the in-situ observations. The point here is to say why we need numerical simulations and that we need to evaluate them and understand their strengths and weaknesses.**

- "P4: L17-20: There are many more recent and informative studies regarding the time scale and decay of the artificial elastic waves. From my experience, your choices of number of subcycles is far too low particularly at 1/12 degree resolution. See for instance: Lemieux et al (2012), Boullion et al (2013), Kimmritz et al (2016), Williams et al (2017)."
  **We added the suggested references. "Note that recent studies (e.g., Lemieux et al., 2012; Bouillon et al., 2013; Williams et al., 2017) showed that more iterations are needed to reach a viscous-plastic (VP) solution. Without doing that, the divergence field will be affected, i.e., being noisy (Dupont, 2017, personal communication). Thus, to what degree it will impact the final averaged ice thickness will vary in space. Such an investigation in the CAA is beyond the scope of this study."**

- "P4 L27: How are the 33km wind fields used to force the simulations with different spatial resolutions? There seems to be many issues which could arise here."
  **We added the text "These forcing fields are linearly interpolated onto model grid". This is done using the NEMO on-the-fly interpolation, which is a standard way to do this in numerical models.**

Interactive
comment

- "P5 L5: It is unclear how the CORE II simulations incorporate the inter-annual variablity of the atmospheric forcing. What is the climatological mean? This deserves more explanation."
  **The CORE-II dataset provides the inter-annual atmospheric fields although with different temporal and spatial resolutions. The climatology of the data set is documented in the reference, Large and Yeager (2009). We understand the CORE-II inter-annual dataset is based on a mixture of NCEP reanalyses and satellite observations with adjustments. This is different from the CGRF (from a GEM simulation) used in other simulations involved in this study. However, the differences between the forcing fields and their impacts are not the focus of this study.**

- "P7 - see 3rd paragraph in general review"
  **An estimate of the skill of the model is needed but very limited time series are available for a fair comparison. Neither the interpolation or the nearest point method is perfect in such comparisons because it is essentially not resolved by such simulations. Thus, we do not think the method used in this study itself affects our results here.**

- "P8 - see 1st and 4th paragraph in general review"
  **First, in our original comparison, we did not address clearly the differences between in-situ observed and simulated ice thickness. This is pointed out by #3 reviewer. The observation (ECCC site data used in this study) represents the "immobile level first-year (seasonal) ice of the uniform thickness that forms close to shore, and is forced by thermodynamic processes". The differences between the observed and simulated ice thickness also explain the reviewer's question on ice thickness at Eureka and Alert.**

- "P8 L25 It's not clear how the data assimilation is taking place. What fields are being assimilated, and in which simulations? How does this affect the results?

What if no assimilation is done?"

**We changed the text to "which is likely due to data assimilation in GLO-RYS2v3" to make it clear. Data assimilation is done only in GLORYS2v3 here, and the technical details of the data assimilation in GLORYS2v3 is documented in the reference, Masina et al., (2015). In their simulation, only the concentration field is assimilated for seaice. Basically, we are including this additional experiment to show that data assimilation can change the model behavior in the region but not necessarily make it closer to observations.**

- "P9 Fig 2: All of the observed timeseries look similar in this figure. Perhaps another figure showing the differences due to location would be useful."
  **Different y-axis scales were used in the plots. The observations were not sampled at the same time, thus interpolation will be involved for the difference-type plot. We tried to keep to the original data as much as possible. Thus, we added "Different y-axis scales are used." in the caption to make it clear. As well, the addition of figure 3 with the mean seasonal cycles, helps highlight the differences.**

- "P12-15: Full timeseries of these fields would be much more interesting to see at these locations rather than seasonal cycles. This would allow us to see if there is a correlation between particular dynamic events and the thermodynamics feedbacks that we expect. Perhaps keep the seasonal cycles as well for completeness."
  **Agree the timeseries without averaging can help to see whether there is any interaction between the two processes but the full timeseries are hard to read on paper unless presented one row for each year. We did have one example at Resolute for 2012 only in our original draft (fig 6 in the old version, and now fig 7 in the revised version). We can add them if the editor think they are worth the space.**

- "P16-20: I do not see what this analysis adds to the story. We already see that there is a seasonal cycles and it must be that daytime melt outweighs nighttime freezing during the melt season."
  **This analysis presents information on the dominant periods of variability (thermodynamic), the lack thereof in terms of the dynamics as well as some detailed information on the details of thermodynamic changes during the break-up period. Thus, we think this material is worth retaining.**

**Answer to specific comments:**

- "P1 L11: A relatively small"
  **Changed.**

- "P1 L22- P2 L4: Confusing, rephrase"
  **Rephrased to "Economically, shipping through the CAA , via the Northwest Passage (NWP), is of particular interest to commercial transport between Europe and Asia because of the great distance savings compared to the current route through the Panama Canal (e.g., Howell et al., 2008; Pizzolato et al., 2016, 2014). This has been a hot topic under the context that Northern Hemisphere sea ice cover has been declining dramatically (e.g., Parkinson et al., 1999; Serreze et al., 2007; Parkinson and Cavalieri, 2008; Stroeve et al., 2008; Comiso et al., 2008; Parkinson and Comiso, 2013), especially after 2007."**

- "P2 L13: Rephrase. Also remove the quotes around statistically significant"
  **The quoted words are from the original reference. We think it is the proper way to cite the original words from a reference. We rephrased the the sentence to "Reduction in the September MYI cover is also found to be -6.4%**

**per decade until 2008 (Howell et al., 2009). But this trend was not "yet sta-
tistically significant" due to the inflow of MYI from the Arctic Ocean, mainly
via the Queen Elizabeth Islands (QEI) gates in August to September (Howell
et al., 2009). With extended data in recent years (until 2016), Mudryk et al.
(2017) showed that the summer MYI decline rate has almost doubled" to
make it clear.**

- "P2 L17: There are"
  **Corrected.**

- "P2 L24: conditions"
  **Changed.**

- "P4 L19 "can" does not make sense here. No-slip boundary conditions define
  that the velocity is zero at the coast line"
  **Removed as requested.**

- " P6 L6: This sentence is unclear and further explanation of the assimilation
  process is required."
  **This sentence has been removed.**

- "P6 L12: delete "only""
  **"only" here is to address the number of observation sites is less in the New
  Icethickness Program compared to the original one. So we prefer to keep
  it.**

- "P6 L13: delete "period""
  **Removed.**

- "P7: L9: calculation"
  **Corrected.**

- "P9 L3: "Cambridge Bay" rather than "the Cambridge Bay""
  **Corrected.**

---

## Author Comment (AC3) · 18 Jan 2018

article graphicx float booktabs [justification=centering]caption

**Thermodynamic and Dynamic Ice Thickness Changes in the Canadian Arctic Archipelago in NEMO-LIM2 Numerical Simulations**

Xianmin Hu, Jingfan Sun, Ting On Chan, and Paul G. Myers

January 18, 2018

**Reply to Reviewer 3:**

**general comments:** "This is an interesting study, comparing sea ice thickness simulations from a numerical model with landfast ice thickness observations at eight sites in the Canadian Arctic Archipelago, separating simulated changes in ice thickness into thermodynamic and dynamic contributions, and describing diurnal oscillations in ice thickness and thermal ice production. However, I feel that the purpose of the work is not clearly articulated. I suggest it could say something like "first, to evaluate the skill of a numerical model in simulating sea ice thickness by comparing the simulations with observations of landfast ice thickness at several sites in the CAA. Two features of the simulations will be then be discussed: 1) the relative importance . . .". I also feel that

the paper does not make sufficiently clear the difference in the properties of the observation data versus the simulation data. The observation data represents immobile level first-year (seasonal) ice of uniform thickness that forms close to shore, and is forced by thermodynamic processes. The simulation data (page 8, line 12) gen- erally represents ice found beyond the near-shore ice and is a mixture of deformed (ridged/rafted) and level first-year ice, young ice and old (perennial) ice, is mobile for part of the year, and is forced by both thermodynamic and dynamic processes. The degree to which we should expect them to agree therefore depends on the concentration of old ice and deformed ice, differences in the timing of freezeup/breakup, etc.

I think that more detail is required to describe the skill of the model. The summary (but not the abstract) mentions the capability of capturing the seasonal cycle and amplitude of ice thickness. This would be clearer if the seasonal cycles were plotted as in Howell et al. (2016). In addition, such a plot would more clearly show the differences/agreement between model results and observations at Resolute and Cambridge Bay. Perhaps the dynamic processes in Figures 4 and 5 could then be used to explain, in part, these differences. Does the model have any significant skill with respect to interannual variability (or does it not, because of snow depth variations on small horizontal scales)?"

**We thank reviewer #3 for the comment about the differences between different types of sea ice, particularly on the site observations ("immobile level first year ice"). This helps us a lot in understanding the discrepancies between the simulated and observed ice thickness. We added the related text both in the data section and the comparison section to state the differences clearly. Please see more in our detailed answers. In the revised version, we added a new section on**

the ice volume budget within different regions in our study area. Comparisons with previous studies are also included to support the inter-annual variability seen in our results.

**minor comments:**

- "Page 1, lines 3-6: "the model captures well the general spatial distribution . . . (âĹij4m and thicker)". While this may be true, the model was compared with land-fast ice thickness observations (first year ice only, no old ice or deformed ice), that are generally not much greater than 2m. Why not describe a general comparison with published data from IceSat, CryoSat or other sources (e.g. Laxon et al., 2013; Tilling et al.,2015), which include the thicker ice types?"
  **We added the related references of ice thickness observations to support our statement. "Here we focus on the ice growth process between December and April of the following year. Figure 4a and 4b show the simulated ice thickness in ANHA12 at the beginning of December and at the end of April, respectively. Geographically, at the end of April, a) very thick sea ice is located in the northern CAA ($\sim$ 4 m by the end of April) with regional maximum ($> 4.5\,m$) at the openings to the Arctic Ocean. This is consistent with the ICESat and Cryosat-2 estimations (e.g., Laxon et al., 2013; Tilling et al., 2015; Kwok and Cunningham, 2015). b) less thick sea ice covers western, and central Parry Channel (just in the west of the site Resolute) and M'Clintock Channel with a thickness of 2.5 m to 3 m. These values are similar to previous obser- vations from airborne electromagnetic surveys (Haas and Howell, 2015) and satellite (Tilling et al., 2017). "**

- "Page 1, lines 6-8: What is meant by "compares well"? Do you mean the seasonal cycles and amplitudes, as stated in the summary? Is agreement with first-year

landfast ice better in the south because there are low concentrations of old ice?"
**To make it clear, the text has been revised to "simulated ice thickness compares reasonably (seasonal cycle and amplitudes) with weekly Environment and Climate Change Canada (ECCC) New Icethickness Program data at first-year landfast ice sites but not at the northern sites with high-concentration of old ice".**

- "Page 1, line 13: Add "at two sites" after "ice fields""
  **Added as suggested.**

- "Page 2, line 34: "this downward trend is mostly associated with changes in snow depth". The meaning of this is not clear. Do you mean that in most cases, the down- ward trend in ice thickness is associated with a positive trend in snow depth (since ice thickness is negatively correlated with snow depth)? Only one of the cases had a significant trend in snow depth, and it was negative, not positive."

  **We revised to "They found statistically significant thinning at the sites except at Resolute, and the detrended inter-annual variability is highly (negative) correlated with snow depth due to insulating effect of the snow (Brown and Cote, 1992)."**

- "Page 3, line 3-4: Change "a sea ice model" to "several sea ice models"?"
  **No. Here the sea ice model is refered to LIM2 sea ice model. The same sea ice model is used for all the simulations included in this study.**

- "Page 6, line 12: Were three of the 11 stations omitted from the analysis because they were on lakes?"
  **Yes. We inserted "The remaining three sites are on lakes (not included in our simulations)." in the revised version.**

- "Page 6, line 16 and elsewhere: The paper would be much easier to read if the full names (not acronyms) were used for the station locations."
**Changed as suggested. We use full names in both table 2 and texts now in the revised version. Acronyms are now used only in figure 1 to keep it concise.**

- "Page 8, line 10: The 3 sites with poor agreement between simulations and observations are in areas with significant concentrations of old ice, while the sites with reasonable agreement are in areas without (see Canadian Ice Service (2011)). Is this the basic reason for the poor agreement at the 3 sites?"
**Yes. We added as suggested. "The sites where the model produced much thicker ice are likely where significant concentration of old ice exists (CIS, 2011)."**

- "Page 8, line 11: I suggest adding a plot of the seasonal cycles of the models and observations (as in Howell et al 2016, Figure 8). This would make it easier to visualize the asymmetric seasonal cycles and summarize the differences in amplitude etc. between the various models."
**Added as suggested as fig 3.**

- "Page 8, line 21: "too thick sea ice". What would be a realistic sea ice thickness, based on the literature, given that there are significant concentrations of old ice in the area?"
**We have revised the texts to "At Eureka, Alert and Alert LT1 sites (Fig. 2 and 3, e, f, and g), there are clear differences between the simulated ice thickness and the observations (âĹij 2 m at Alert/Alert LT1 and âĹij 1 m at Eureka). Note neither ANHA4 or ANHA12 has the capability to resolve the difference between Alert and Alert LT1, thus, the same simulated values are shown on the figure for both sites. The differences between simulations and observations could be an initial value problem, particularly at**

**Eureka (Fig. 2g). However, given high concentrations of old ice are at these sites, observations represent the immobile level first-year ice only. Thus, the model and the observations may not be representing the same type of ice.**

- "Page 10, line 6-7: The meaning isn't clear. "Thus, it is likely due to another physical process such as advection from surrounding areas" (?)"
**Changed as suggested.**

- "Page 17: I suggest reversing the order of Figure 8 and 9, so that they are in the same order as in the text."
**Changed as suggested.**

**Answer to minor comments:**

- "Page 1, Line 21: "overturning""
**Corrected.**

- "Page 2, line 17: "there are still""
**Corrected.**

- "Page 2, line 30: "evaluated the""
**Corrected.**

- "Page 4; Table 1: "subcycling" (?)"
**Corrected.**

- "Page 6, lines 16 and Table 2: Change "Carol" to "Coral"."
**Corrected.**

[Figure]

- "Page 8, line 19: Add "(Fig. 2c and d)"."
  **Added as suggested.**

- "Page 8, line 20: Change "MEU" to "WEU"."
  **Corrected to full name "Eureka".**

- "Page 8, line 24: "green line" (add space)"
  **Corrected.**

- "Page 8, line 34: Add ".""
  **Corrected.**

- "Page 10, line 21: "just south of the site YRB" (?)"
  **Changed to "just to the west of the site Resolute".**

- "Page 12, line 4: "spatial""
  **Corrected.**

- "Page 16, line 9: "supports the notion that" (?)"
  **Changed as suggested.**

- "Page 19, line 4: "constraints""
  **Corrected.**

**Supplement:**

[revised manuscript text omitted]

---

## Author Response (AR2)

**Thermodynamic and dynamic ice thickness contributions in the Canadian Arctic Archipelago in NEMO-LIM2 numerical simulations**

Xianmin Hu, Jingfan Sun, Ting On Chan, and Paul G. Myers

March 16, 2018

Dear Christian,

Thanks for your suggestions. We have revised the manuscript based on the reviewers' comments. Detailed replies are provided later. Basically, we removed wavelet analysis and related text, revised the summary and discussion section. One of the reason that polynyas were not well produced in our simulations is tides were not included in the simulations. In addition, 1/12 degree resolution can't resolve the fjords discussed in Melling et al. (2015). Thus, I don't have much confidence in the simulation on this aspect. We included the explanation in summary and discussion section in current draft.

Regards,
Xianmin

**Reply to reviewer 1:**
We thank reviewer 1 for the comments that help to further improve the manuscript. Our detailed responses are given as follows.

"1. After reading this manuscript again, I think section 3.2.3 does not need to be included. It is confusing and considering the new section 3.3 the manuscript is already quite long. Another reviewer also suggested it is not needed and I strongly agree."
**We removed this section and all wavelet related texts. Please find the details in the diff file.**
"2. Section 4 does not present solid conclusions about integrating what was learned and what needs to be done with respect to modeling in the CAA. Figure 18 and its associated discussion is way out of place in the Summary and Discussion at the end as it is never a good idea to introduce a new figure at the end of the manuscript. In fact, everything after Line 4 on Page 27 are a laundry list of limitations. A better approach to Section 4 would be to integrate the results with these limitations rather than just list the problems with models in general."

**We revised Section 4. The new version is**

**"Increasing model horizontal resolution does not result in much noticeable change/improvement in sea ice thickness simulation, at least between $1/4°$ and $1/12°$. We presented the sea ice thickness simulated within the CAA with a relatively simple (not multi-category) sea ice model, LIM2, with both $1/4°$ and $1/12°$ resolutions from 2002 to 2016. Simulations can capture the ice thickness asymmetric seasonal cycle and amplitude, and compares reasonably well with the ECCC observations at most sites. In general, the difference is not visible between the runs with different horizontal resolutions. We expect model resolution to play a big role when it resolves much smaller scale, e.g., sufficient to resolve a ridge/lead. ANHA12 does show differences at Eureka (Fig. 2g), but this is not related to the sea ice model physics but improvements in the local coastline, and thus the regional circulation and the dynamic component (not shown). On this aspect, our simulations do not have enough resolution to resolve the fjord process, which is important to study the hidden polynyas in Melling et al. (2015). This study focuses on the large scale features, e.g., the simulations can produce reasonable spatial distribution of the thickness (very thick ice in the northern CAA, thick ice in the west-central Parry Channel and thin ice in the eastern and southern regions of CAA).**

**The dynamic contribution should be considered in the off-shore ice growth and basin scale ice volume budget in the CAA. We studied the spatial distribution of the thermodynamic and dynamic ice growth in winter months. Relatively smaller thermodynamic contribution in the winter season is found in the thick ice covered areas, with larger contributions in the thin ice covered regions. Large dynamic ice growth is found along the northern CAA coast, the west of Bank Island, Byam Martin Channel, and northwest Baffin Bay region (e.g., Smith Sound, Jones Sound and Lancaster Sound). On the basin scale, the inter-annual variations of the winter maximum ice volume in the northern CAA, Parry Channel and Baffin Bay are controlled by the thermodynamic growth and lateral transport. While both components demonstrate significant inter-annual variabilities, there is no clear trend in the winter maximum ice volume within the northern CAA and Baffin Bay regions but a downward trend ($r^2 \approx 0.6$) in Parry Channel region. In the northern CAA, the lateral transport is mainly through the northern gates and Byam Martin Channel but large ice volume flux could also flow south via Penny Strait when there is large**

inflow through the northern gates. Ice flow via Byam Martin Channel into Parry Channel is balanced by outflow into M'Clintock Channel on average. Eastward sea ice export through Lancaster Sound mouth is a big term in Parry Channel ice volume budget, but is much smaller than the influx from Smith Sound and outflux through Davis Strait in the ice volume budget of Baffin Bay. These estimates are comparable to limited available studies, however, further evaluations are still in need to confirm the quantities and variations.

It should be noted that landfast ice parameterizations and tides were not included in our simulations. The sea ice model utilized here does produce zero-motion sea ice (e.g., Fig. 4d), however, more realistic physical parameterizations (Lemieux et al., 2016) are not applied in our simulations yet. With such parameterizations, we expect great improvements in simulating the widely existing landfast ice in the CAA region (Melling, 2002; Galley et al., 2012; Haas and Howell, 2015; Howell et al., 2016). Below the ice, tidal current plays an important role in the formation of open and hidden polynyas by enhancing mixing, bringing warm subsurface water towards surface or into fjords over the sills (Melling et al., 2015). Luneva et al. (2015) also showed there is much larger tidal impact on ice thickness in the CAA than the Arctic Ocean in numerical sensitivity experiments. Unfortunately, tides are not included in the ocean component of our current simulations. Thus, polynyas, the important features in this region, were not well produced in the simulations. Neither can we study the detailed realistic physical formation processes proposed by Melling et al. (2015)."

Note we moved the texts about CORE-II snowfall issue to the result, section 3.1.
" At Alert/Alert LT1, both ANHA4-CGRF (blue line) and ANHA12-CGRF (red line) show similar inter-annual trends to that in GLO-RYS2v3 (which extends back to 1993, green line), meaning it is likely a pure initial value problem rather than the model equilibrium issue mentioned in Howell et al. (2016). In addition, the seasonal cycle is not clear in the GLORYS2v3 product. The issue is also present in some years, i.e., 2005–2007, in the ANHA4-CGRF and ANHA12-CGRF simulations. ANHA4-CORE (orange line) is generally improved compared with the observations in both the amplitude and seasonal cycle. However, this improvement was achieved by accident, and is related to a snow depth issue in this simulation. The snowfall data from CORE-II has a monthly resolution, which is possibly too coarse temporally (Hayashida, personal communication, 2017). This leads the snow depth to drop to close to zero quickly during the the first year of the simulation but not in the CGRF simulations with hourly snowfall. Thus, it does not indicate that CORE-II forcing is performing better than other atmospheric forcing datasets in this

**region.”**

**Reply to reviewer 3:**

We thank reviewer 3 again for the comments. They have greatly improved the manuscript. Here are our detailed responses.

**Minor comments**

“Page 1, line 17: No significant variation . . . . Do you mean No significant trend as on page 26, line 20? A variation or shrinkage of as much as 1/3 is noted on page 22, line 9.”
**We replaced "variation" with "trend".**
“Page 1, line 18: The two main contributors balance each other . If they balanced each other, there would not be a shrinkage of 1/3 in ice volume, though admittedly the interannual variability of the individual contributors is larger than that of ice volume. I suggest something like the two main contributors (. . . ) have high interannual variabilities which largely balance each other, so that maximum ice volume varies interannually by about x%. Further quantitative evaluation is required .”
**We revised the sentence as suggested.**
**The new version is " The two main contributors (thermodynamic growth and lateral transport) have high inter-annual variabilities which largely balance each other, so that maximum ice volume can vary interannually by $\pm 12\%$ in the northern CAA, $\pm 15\%$ in Parry Channel, and $\pm 9\%$ in Baffin Bay, respectively. Further quantitative evaluation is required.”**

**Technical comments:**

- “Pg 1, line 2: framework”
  **Corrected.**

- “Pg 1, line 9: high concentrations”
  **Corrected.**

- “Pg 9, line 12: concentrations”
  **Corrected.**

- “Pg 9, line 29: same type”
  **Corrected.**

- “Pg 11, Fig. 3d: remove 2016 from x axis.”
  **Corrected.**

- "Pg. 25, line 8: Sou and Flato (2009)"
  **Corrected**

- "Pg. 25, line 17: Sou and Flato (2009) and Terwisscha van Scheltinga et al. (2010)"
  **Corrected**

- "Pg. 25, line 19: much more reasonable"
  **Corrected**

- "Pg. 29, line 14: framework of"
  **Corrected**

**Thermodynamic and dynamic ice thickness contributions in the Canadian Arctic Archipelago in NEMO-LIM2 numerical simulations**

Xianmin Hu[1,+], Jingfan Sun[1,++], Ting On Chan[1,+++], and Paul G. Myers[1]

[1]Department of Earth and Atmospheric Sciences, University of Alberta, Edmonton, T6G 2E3, Canada
[+]now at Bedford Institute of Oceanography, Fisheries and Oceans Canada, Dartmouth, Nova Scotia, Canada
[++]summer intern from Zhejiang University, 38 Zheda Road, Hangzhou, China, 310027
[+++]now at Skytech Solutions Ltd., Canada

*Correspondence to:* Xianmin Hu(xianmin@ualberta.ca)

**Abstract.** Sea ice thickness evolution within the Canadian Arctic Archipelago (CAA) is of great interest to science, as well as local communities and their economy. In this study, based on the NEMO numerical  framework including the LIM2 sea ice module, simulations at both $1/4°$ and $1/12°$ horizontal resolution were conducted from 2002 to 2016. The model captures well the general spatial distribution of ice thickness in the CAA region, with very thick sea ice ($\sim 4\,m$ and thicker) in the northern CAA, thick sea ice ($2.5\,m$ to $3\,m$) in the west-central Parry Channel and M'Clintock Channel, and thin ($< 2\,m$) ice (in winter months) on the east side of CAA (e.g., eastern Parry Channel, Baffin Islands coast) and in the channels in southern areas. Even though the configurations still have resolution limitations in resolving the exact observation sites, simulated ice thickness compares reasonably (seasonal cycle and amplitudes) with weekly Environment and Climate Change Canada (ECCC) New Icethickness Program data at first-year landfast ice sites except at the northern sites with  high-concentration of old ice. At $1/4°$ to $1/12°$ scale, model resolution does not play a significant role in the sea ice simulation except to improve local dynamics because of better coastline representation. Sea ice growth is decomposed into thermodynamic and dynamic (including all non-thermodynamic processes in the model) contributions to study the ice thickness evolution. Relatively smaller thermodynamic contribution to ice growth between December and the following April is found in the thick and very thick ice regions, with larger contributions in the thin ice covered region.  No significant trend in winter maximum ice volume is found in the northern CAA and Baffin Bay while a decline ($r^2 \approx 0.6$, $p < 0.01$) is simulated in Parry Channel region. The two main contributors (thermodynamic growth and lateral transport) have high inter-annual variabilities which largely balance each other, so that maximum ice volume can vary interannually by $\pm 12\%$ in the northern CAA, $\pm 15\%$ in Parry Channel, and $\pm 9\%$ in Baffin Bay, respectively. 
[revised manuscript text omitted]

15   distribution of the thickness (very thick ice in the northern CAA, thick ice in the west-central Parry Channel and thin ice in the eastern and southern regions of CAA).

    The dynamic contribution should be considered in the off-shore ice growth and basin scale ice volume budget in the CAA. We studied the spatial distribution of the thermodynamic and dynamic  ice growth in winter months. Relatively smaller thermodynamic contribution in the winter season is found in the thick ice covered areas, with larger contributions in

20   the thin ice covered regions

 . Large dynamic ice growth is found along the northern CAA coast, the west of Bank Island, Byam Martin Channel, and northwest Baffin Bay region (e.g., Smith Sound, Jones Sound and Lancaster Sound). On the basin scale, the inter-annual variations of the winter maximum ice volume in the northern CAA, Parry Channel and Baffin Bay are controlled by the thermodynamic growth and lateral transport. While both components demonstrate significant inter-annual variabilities, there is no clear trend in the winter maximum ice volume within the northern CAA and Baffin Bay regions but a downward trend ($r^2 \approx 0.6$) in Parry Channel region. In the northern CAA, the lateral transport is mainly through the northern gates and Byam Martin Channel but large ice volume flux could also flow south via Penny Strait when there is large inflow through the northern gates. Ice flow via Byam Martin Channel into Parry Channel is balanced by outflow into M'Clintock Channel on average. Eastward sea ice export through Lancaster Sound mouth is a big term in Parry Channel ice volume budget, but is much smaller than the influx from Smith Sound and outflow through Davis Strait in the ice volume budget of Baffin Bay. These estimates are comparable to limited available studies, however, further evaluations are still in need to confirm the quantities and variations.

 It should be noted that landfast ice parameterizations and tides were not included in our simulations. The sea ice model utilized here does produce zero-motion sea ice (e.g., Fig. 4d), however, more realistic physical parameterizations (e.g., Lemieux et al., 2016) are not applied in our simulations yet.  With such parameterizations, we expect great improvements in simulating the widely existing landfast ice in the CAA region (Melling, 2002; Galley et al., 2012; Haas and Howell, 2015; Howell et al., 2016) . Below the ice, tidal current plays an important role in the

~~Another ice model physics related process is the snow depth at the surface. Brown and Cote (1992) , Dumas et al. (2006) and Howell et al. (2016) pointed out that the snow thickness plays a major role in a sea ice thickness simulation. The snowfall data from CORE-II has a monthly resolution, which is possibly too coarse temporally. This leads the snow depth to drop to close to zero quickly during the the first year of the simulation. This never happens with the hourly CGRF forcing, as well as other datasets with daily snowfall (Hayashida, 2017) .~~

~~A large source of bias is related to the radiation fluxes (shortwave and longwave) at the atmosphere-ocean interface. The CGRF product use the GEWEX correction (more details in Smith et al. (2014) ) to minimize the bias in the original output from the atmospheric model. However, it was found that this correction is not needed for years after 2011 (Paquin, 2017) . A test run was done to estimate the influence of this correction. A large impact was seen in the Arctic Ocean but not over most of the CAA (not shown) except the north. Without the GEWEX correction, the model tends to simulate thinner ice with a thinner snow depth at the same time, e.g., at Eureka (Fig. ??). The impact is smaller at Alert and Alert LT1 and negligible for the rest of the ECCC sites used in this study. Further investigation is undergoing.~~

zero. Another factor that may affect the local ice thickness is the tides (Luneva et al., 2015) , which is not included formation of open and hidden polynyas by enhancing mixing, bringing warm subsurface water towards surface or into fjords over the sills (Melling et al., 2015) . Luneva et al. (2015) also showed there is much larger tidal impact on ice thickness in the CAA than the Arctic Ocean in numerical sensitivity experiments. Unfortunately, tides are not included in the ocean component of our current

5    simulations. Thus, polynyas, the important features in this region, were not well produced in the simulationsused in this study as well. All these factors will be considered in the future studies. Neither can we study the detailed realistic physical formation processes proposed by Melling et al. (2015) .

*Data availability.* For access to the model data contact P.G. Myers (pmyers@ualberta.ca)

*Competing interests.* NONE

*Acknowledgements.* For access to the model data contact P.G. Myers (pmyers@ualberta.ca). We gratefully acknowledge the financial and logistic support of grants from the Natural Sciences and Engineering Research Council (NSERC) of Canada. These include a Discovery Grant
5   (rgpin 227438-09) awarded to P.G.M., Climate Change and Atmospheric Research Grants (VITALS - RGPCC 433898 and the Canadian Arctic Geotraces program - RGPCC 433848), and Polar Knowledge (432295). We are grateful to the NEMO development team and the Drakkar project for providing the model and continuous guidance, and to Westgrid and Compute Canada for computational resources. We also thank G. Smith for the CGRF forcing fields that made available by Environment and Climate Change Canada. We appreciate the Environment and Climate Change Canada New Arctic Ice Thickness Program for providing valuable sea ice thickness measurements used in this study.
10   Greenland freshwater flux data analyzed in this study is that presented in Bamber et al. (2012) and is available on request as a gridded product. We thank NCAR/UCAR for making Dai and Trenberth Global River Flow and Continental Discharge Dataset available. We acknowledge WCRP/CLIVAR Ocean Model Development Panel (OMDP) for sponsoring and organizing the Coordinated Ocean-sea ice Reference Experiments dataset (CORE). We also acknowledge Mercator Ocean for providing the GLORYS model output for initial and open boundary conditions. The GLORYS reanalysis project is carried out in the framework of the European Copernicus Marine Environment
15   Monitoring Service (CMEMS). We also appreciate the three anonymous reviewers for their valuable comments, which have greatly improved this manuscript.

[revised manuscript text omitted]